# scRL: Utilizing Reinforcement Learning to Evaluate Fate Decisions in Single-Cell Data

**DOI:** 10.3390/biology14060679

**Published:** 2025-06-11

**Authors:** Zeyu Fu, Chunlin Chen, Song Wang, Junping Wang, Shilei Chen

**Affiliations:** 1State Key Laboratory of Trauma and Chemical Poisoning, Institute of Combined Injury, Chongqing Engineering Research Center for Nanomedicine, College of Preventive Medicine, Army Medical University, Chongqing 400038, China; fuzeyu99@126.com (Z.F.); swang1981@tmmu.edu.cn (S.W.); 2Department of Rehabilitation Medicine, The First Affiliated Hospital, Sun Yat-sen University, Guangzhou 510080, China; chenchlin3@mail2.sysu.edu.cn

**Keywords:** single–cell, reinforcement learning, actor–critic, fate decisions, trajectory inference, dimensionality reduction

## Abstract

Understanding how cells develop into different types during growth and disease is crucial for advancing medicine, but current methods cannot pinpoint exactly when and where cells make these critical decisions. We developed a new artificial intelligence tool called single-cell reinforcement learning that treats cell development like a strategic decision-making game. Just as a chess player learns to make optimal moves, our system learns to identify the precise moments when cells decide their future fate—whether to become blood cells, immune cells or other specialized types. We tested this approach on various biological systems, including normal human blood cell development, cancer cells, mouse organ development and cells responding to radiation damage. Our method consistently outperformed fifteen existing state-of-the-art tools and successfully identified early decision points that occur before cells show obvious signs of commitment to specific lineages. Additionally, we discovered previously unknown regulatory factors that control these decisions. This breakthrough provides scientists with a powerful new way to understand how cells make developmental choices, which could lead to better treatments for diseases like cancer and improved strategies for regenerative medicine. By revealing the hidden decision-making logic of cellular development, this work opens new possibilities for controlling and directing cell fate in therapeutic applications.

## 1. Introduction

### 1.1. Single-Cell Sequencing and Dimensionality Reduction

Single-cell sequencing now enables genome-wide measurements at the resolution of individual cells, exposing previously hidden heterogeneity across tissues and disease states [1,2,3]. Each experiment routinely profiles tens of thousands of genes in thousands to millions of cells, yielding data that are not only high-dimensional but also sparse, noisy and burdened by technical artefacts [4]. Consequently, DR (dimensionality-reduction) methods have become indispensable for extracting biological structure from such data [5].

A major downstream application of DR in the single-cell field is pseudotime analysis, which seeks to order cells along putative developmental trajectories [6,7]. Visualization techniques—including PCA (principal component analysis), t-SNE (t-distributed Stochastic Neighbour Embedding) and UMAP (Uniform Manifold Approximation and Projection)—facilitate exploratory analysis and often serve as the basis for trajectory inference [8,9,10]. Indeed, tools such as PAGA, Monocle 3, Slingshot, and CAPITAL exploit UMAP embeddings to reconstruct branching lineages [11,12,13,14]. Yet, despite their success in ordering cells, existing approaches seldom address *where* and *when* lineage commitment occurs, nor do they provide a quantitative framework to evaluate these fate decisions. In reality, differentiation is not strictly hierarchical but resembles a rugged landscape in which progenitors become lineage-restricted at context-dependent times [15]. Detecting the continuous states of fate decisions that guide this process therefore remains an open problem.

### 1.2. Manifold Learning for Dimensionality Reduction

Linear DR techniques, exemplified by PCA, are computationally efficient but fail to capture the nonlinear geometry typical of biological systems [16]. Manifold-learning algorithms address this limitation by assuming that high-dimensional observations reside on, or near, a low-dimensional manifold embedded in ambient space [10,17]. Isomap [18], LLE (Locally Linear Embedding) [19], t-SNE [20] and UMAP [10] each preserve different aspects of the manifold—geodesic distances, local linear reconstructions or neighbourhood probabilities—thereby recovering structure inaccessible to linear projections. Theoretical foundations draw on differential geometry, topology and graph theories, providing a flexible toolkit for biological data characterized by nonlinear relationships and multiple developmental branches [16].

### 1.3. Applications in Single-Cell Data Analysis

Manifold learning has become integral to single-cell workflows because it reveals continuous developmental paths, rare subpopulations and functional gradients that remain hidden in the original space [21,22]. UMAP visualizations, for example, readily separate major immune lineages and their subtypes from ${\mathrm{CD45}}^{+}$ scRNA-seq profiles [23]. Specialized adaptations align transcriptomic with epigenomic or electrophysiological modalities [24,25], and Gaussian-process latent variable models capture noisy gene expression as smooth functions of latent states [26]. Nevertheless, even state-of-the-art manifold approaches primarily *describe* trajectories; they do not explicitly *evaluate* the decisions that drive differentiation.

### 1.4. Reinforcement Learning for Cellular Differentiation Trajectories

RL (reinforcement learning) offers a principled framework for sequential decision-making in complex, dynamic environments [27]. An RL agent iteratively selects actions, observes transitions and updates its policy to maximize the cumulative reward—attributes that have delivered breakthroughs in robotics, game playing and navigation [28,29,30]. Conceptually, cellular differentiation mirrors this paradigm: (i) a transcriptomic profile represents the *state*, (ii) regulatory events such as gene activation or repression correspond to *actions*, (iii) differentiation steps define *state transitions*, and (iv) developmental constraints or lineage markers provide *rewards* [31].

RL is therefore well-suited to explore high-dimensional single-cell spaces, balance the discovery of novel routes with optimization of known trajectories and pinpoint branch points where fate decisions occur [32,33]. Deep RL extends these capabilities to raw, high-dimensional inputs by coupling neural networks with value- or policy-based optimization [34]. For scRNA-seq data—often exceeding 10,000 genes—such expressivity is essential [35]. Framing differentiation as an RL task promises to reveal regulatory logic that conventional pseudotime tools overlook, particularly in multi-lineage systems with complex branching topologies [36].

### 1.5. Our Contribution

Here we introduce scRL, a reinforcement-learning framework that integrates manifold learning, an actor–critic architecture and biologically informed reward functions to decode fate decisions from single-cell transcriptomes. Briefly, we (i) construct an interpretable latent space via LDA, (ii) embed this space onto a two-dimensional grid, preserving UMAP topology, and (iii) train an RL agent whose critic learns state values reflecting lineage potential. scRL thereby identifies pre-expression decision states, quantifies lineage and gene decision intensities, and uncovers regulatory factors. Across diverse datasets—including human hematopoiesis, acute myeloid leukemia, mouse endocrinogenesis, *Dapp1* knockout and irradiation injury—scRL outperformed benchmark fate-inference and pseudotime methods, revealed novel regulators and mapped dynamic fate biases (see Results, Figure 1, Figure 2, Figure 3, Figure 4, Figure 5, Figure 6, Figure 7, Figure 8, Figure 9 and Figure 10). By viewing differentiation through the lens of sequential decision optimization, scRL provides a rigorous, extensible framework for studying cell-fate decisions, with potential applications in regenerative medicine, oncology and developmental biology.

## 2. Materials and Methods

### 2.1. Architecture and Workflow of scRL

scRL couples single-cell manifold learning with an actor–critic algorithm in three sequential stages (Figure 1).
Stage 1: Preprocessing and grid embedding (Figure 1A)(i)Dimension reduction. The gene count matrix is first projected into a latent space.(ii)Manifold construction and clustering. A UMAP embedding is built from the latent space matrix and Leiden clustering is applied to annotate coarse cell populations.(iii)Grid embedding. To obtain a lattice that preserves local topology yet yields fully connected paths, edges in the UMAP k-NN graph are filtered by a Canny-style detector; surviving edges are rasterized into an m×n grid. This transforms scattered points into a continuum that spans primitive to terminal states.(iv)Pseudotime assignment. Dijkstra’s shortest-path algorithm is run on the grid graph from a user-specified root cluster to derive a monotonic pseudotime score for every grid node.(v)Bidirectional projection. Each cell inherits the coordinates, pseudotime and neighbourhood of its nearest grid node; conversely, grid nodes maintain links back to the original transcriptomic space.
Figure 1The comprehensive architecture of the scRL framework for cellular fate decision analysis. (**A**) The preprocessing pipeline, which integrates dimensionality reduction, clustering, an edge detection-based grid embedding algorithm, subpopulation mapping, pseudotime computation via Dijkstra’s algorithm and bidirectional projection between embedding spaces. Gene expression data and clustering labels form the basis for constructing reward environments, which operate in two modes: a decision mode (reward diminishing along pseudotime) and a contribution mode (reward intensifying along pseudotime). (**B**) Core functional modules comprising a gene module (gene potential and decision values), a lineage module (lineage potential and decision values) and a trajectory module (differentiation trajectory analysis). (**C**) The actor–critic reinforcement learning architecture, implemented as distinct neural network architectures within scRL. The critic (value network) processes the latent space—which we validated to be optimally represented by LDA for robust feature extraction—to output cell-state value, specifically the decision or contribution intensity. Concurrently, the actor (policy network) learns and outputs the optimal differentiation trajectories within the grid environment.
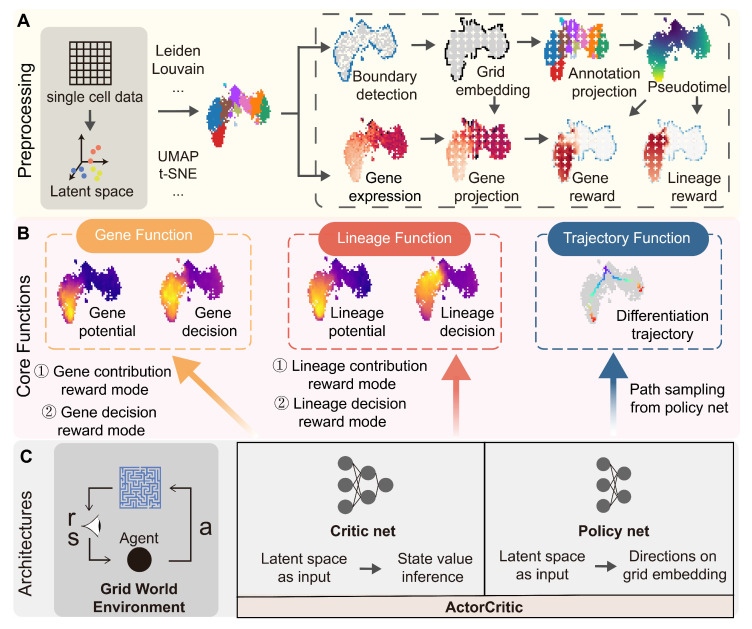

Stage 2: Reward design and module definition (Figure 1B)

Gene expression levels create two complementary reward landscapes:Decision mode—the reward decays exponentially with pseudotime, emphasizing early contribution signals.Contribution mode—the reward increases with pseudotime, capturing cumulative lineage output.

These landscapes feed the following three functional modules:Gene module calculates gene-level potential and decision intensity.Lineage module outputs lineage potential and decision intensity.Trajectory module simulates differentiation by rolling out the learned policy on the grid.

Definition of Decision and Contribution Intensity

Decision Intensity quantifies the “fate intensity” or “decision value” of a cell state, particularly focusing on its significance in shaping future fate decisions *before* overt lineage commitment. It is derived from the state value learned by the critic when operating in the “decision mode,” where rewards are structured to emphasize early signals by decaying exponentially along pseudotime. Higher decision intensity therefore indicates a cell state with greater potential to influence downstream lineage choices, capturing the “pre-expression decision states” where crucial developmental commitments are made.Contribution Intensity quantifies the cumulative “lineage output” or “lineage contribution” from a given cell state over time. It is derived from the state value learned by the critic when operating in the “contribution mode,” where rewards are structured to emphasize cumulative output by increasing along pseudotime. Higher contribution intensity reflects a cell’s accumulated propensity or success in differentiating towards and populating a specific lineage over the course of its developmental trajectory.

Stage 3: Actor–critic learning (Figure 1C)

Critic (value network). A multilayer perceptron ingests latent coordinates and predicts the state value, V(s), for each grid node.Actor (policy network). A parallel network outputs a stochastic policy, π(a|s), that favors moves increasing expected cumulative reward.Optimization. Networks are updated with an advantage formulation and an adaptive learning rate until both cumulative reward and maximal V(s) converge.

Validation workflow (Figure A1)

We benchmarked scRL on human ${\mathrm{CD34}}^{+}$ haematopoietic cells (S-SUBS8) and mouse endocrinogenesis (GSE132188). Lineage-specific markers—*GATA1*, *IRF8*, *EBF1* for hematopoiesis; *Ngn3*, *Fev* for endocrinogenesis—served as external validators of pre-expression decision states. Cluster-specific rewards were assigned to erythroid, myeloid and lymphoid (hematopoiesis) or early/late endocrine trajectories. After training

(a)State values were back-projected onto the UMAP to visualize decision hotspots;(b)Training dynamics showed smooth convergence of the cumulative reward and stabilization of the maximum state value, confirming learning efficacy;(c)Tabular Q-learning on discretized state spaces reproduced the neural results, validating the actor–critic implementation.

### 2.2. Grid-Based Embedding Representation

To transform two-dimensional cell embeddings into a structured grid representation that preserves topological relationships while enabling systematic trajectory analysis, we developed a grid mapping algorithm that discretizes the continuous embedding space into a regular lattice structure.

Grid Generation: Given a two-dimensional embedding matrix, $X\in R^{m\times 2}$, representing *m* cells, we first identify the spatial boundaries:(1)$$\begin{array}{cc} x_{\mathrm{right}} & =X\left[\arg\max_{i}X_{i,1},:\right],\;x_{\mathrm{left}}=X\left[\arg\min_{i}X_{i,1},:\right] \end{array}$$(2)$$\begin{array}{cc} x_{\mathrm{top}} & =X\left[\arg\max_{i}X_{i,2},:\right],\;x_{\mathrm{bottom}}=X\left[\arg\min_{i}X_{i,2},:\right] \end{array}$$

We then generate an n×n regular grid spanning the embedding space:(3)$$\begin{array}{cc} x_{k} & =x_{\mathrm{left}}+\frac{k-1}{n-1}\left(x_{\mathrm{right}}-x_{\mathrm{left}}\right),\;k=1,2,\ldots ,n \end{array}$$(4)$$\begin{array}{cc} y_{l} & =y_{\mathrm{bottom}}+\frac{l-1}{n-1}\left(y_{\mathrm{top}}-y_{\mathrm{bottom}}\right),\;l=1,2,\ldots ,n \end{array}$$

The grid points are defined as $G=\{\left(x_{k},y_{l}\right):k,l=1,2,\ldots ,n\}$, yielding $n^{2}$ total grid positions.

Boundary Detection: To identify the data boundary, we compute angular distances from spine points to all cells. For each spine point $s_{i}$, we calculate:(5)$$\begin{array}{cc} \theta_{i,j} & =\arctan\left(\frac{s_{i,x}-X_{j,x}+\epsilon }{s_{i,y}-X_{j,y}+\epsilon }\right) \end{array}$$(6)$$\begin{array}{cc} d_{i,j} & =\left|\right|s_{i}-X_{j}{\left|\right|}_{2} \end{array}$$
where $\epsilon =10^{-6}$ prevents division by zero. Boundary cells are identified as:(7)$$B_{i}=\left\{\arg\max_{j:\theta_{i,j}=\theta }d_{i,j}:\theta \in \Theta_{i}\right\}$$
where $\Theta_{i}$ represents the set of unique angular values from spine point $s_{i}$.

Grid Masking: We apply a masking procedure to remove grid points in regions with insufficient cell density. Using *j* observer points, we compute angular distances and identify valid grid regions:(8)$$M_{i}=\left\{g\in G:\mathrm{rank}\left(d_{i,g}\right)>\max\left(\mathrm{rank}\left(d_{i,b}\right)\right)+j,\forall b\in B\right\}$$
where $\mathrm{rank}(d_{i,g})$ denotes the distance rank of grid point *g* from observer *i*.

Adjacency Construction: For the remaining mapped grid points, $G_{\mathrm{mapped}}$, we construct an 8-connectivity adjacency matrix, A:(9)$$A_{g_{1},g_{2}}=\left\{\begin{array}{cc} 1 & \mathrm{if}\;||pos\left(g_{1}\right)-pos\left(g_{2}\right){\left|\right|}_{\infty }=1\\ 0 & \mathrm{otherwise} \end{array}\right.$$
where pos(g) returns the (i,j) grid coordinates of grid point *g*.

Boundary Refinement: Grid boundary points are identified as those with fewer than 8 neighbors:(10)$$G_{\mathrm{boundary}}=\left\{g\in G_{\mathrm{mapped}}:\sum_{g^{\prime }}A_{g,g^{\prime }}<8\right\}$$

A pruning step ensures graph connectivity by iteratively adding adjacent points to isolated boundary regions, maintaining a single connected component in the final grid graph $G=(G_{\mathrm{mapped}},A)$.

### 2.3. Pseudotime Alignment via Graph-Based Distance Computation

To establish a coherent temporal ordering across the grid representation, we developed a pseudotime alignment algorithm that leverages graph-based shortest path computation to propagate temporal information from user-defined starting points throughout the grid network.

Starting Point Selection: The algorithm supports two initialization modes. For single-cell initialization with a specified early cell, $c_{0}$, we identify the corresponding grid point:  (11)$$g_{\mathrm{start}}=\arg\min_{g\in G_{\mathrm{mapped}}}\left|\right|X_{c_{0}}-G_{g}{\left|\right|}_{2}$$

For cluster-based initialization with an early cluster, $C_{\mathrm{early}}$, we sample $n_{s}$ starting points from the intersection of cluster and boundary grids:(12)$$S={\mathrm{sample}}_{n_{s}}\left(\left\{g\in G_{\mathrm{mapped}}:\mathrm{cluster}\left(g\right)=C_{\mathrm{early}}\right\}\cap G_{\mathrm{boundary}}\right)$$
where $G_{\mathrm{boundary}}$ represents boundary grid points if the boundary constraint is enabled.

Graph Construction: We construct an undirected graph, H=(V,E), from the grid adjacency matrix:(13)$$V=G_{\mathrm{mapped}},\;E=\left\{\left(g_{i},g_{j}\right):A_{g_{i},g_{j}}=1\right\}$$
where A is the 8-connectivity adjacency matrix from the grid construction phase.

Single-Component Pseudotime Calculation: For a connected graph, we apply Dijkstra’s algorithm from each starting point, s∈S:(14)$$d_{s}\left(g\right)=\mathrm{Dijkstra}\left(H,s,g\right)\;\forall g\in V$$

The mean pseudotime across all starting points is computed as:(15)$$\bar{d}\left(g\right)=\frac{1}{\left|S\right|}\sum_{s\in S}d_{s}\left(g\right)$$

Multi-Component Handling: When the graph contains multiple connected components $\left\{C_{1},C_{2},\ldots ,C_{k}\right\}$, we identify the main component containing the majority of starting points:(16)$$C_{\mathrm{main}}=\arg\max_{i}\left|S\cap C_{i}\right|$$

For each secondary component, $C_{j}$, where j≠main, we establish connection through the closest grid pair:(17)$$\left(g_{\mathrm{main}},g_{j}\right)=\arg\min_{g_{m}\in C_{\mathrm{main}},g_{s}\in C_{j}}\left|\right|G_{g_{m}}-G_{g_{s}}{\left|\right|}_{2}$$

The pseudotime offset for component $C_{j}$ is determined by:(18)$$\tau_{j}=\bar{d}\left(g_{\mathrm{main}}\right)$$

Pseudotime within secondary component $C_{j}$ is calculated as:(19)$$d_{j}\left(g\right)=\mathrm{Dijkstra}\left(H{|}_{C_{j}},g_{j},g\right)+\tau_{j}\;\forall g\in C_{j}$$

Normalization: The final pseudotime values are normalized to the unit interval:(20)$$\psi \left(g\right)=\frac{d\left(g\right)-\min_{g^{\prime }\in V}d\left(g^{\prime }\right)}{\max_{g^{\prime }\in V}d\left(g^{\prime }\right)-\min_{g^{\prime }\in V}d\left(g^{\prime }\right)}$$
where d(g) represents the unified distance function across all components:(21)$$d\left(g\right)=\left\{\begin{array}{cc} \bar{d}\left(g\right) & \mathrm{if}\;g\in C_{\mathrm{main}}\\ d_{j}\left(g\right) & \mathrm{if}\;g\in C_{j},j\neq \mathrm{main} \end{array}\right.$$

### 2.4. Bidirectional Projection Between Cells and Grid

To establish correspondence between the original cell embedding and the grid representation, we developed bidirectional projection methods that enable transfer of annotations and continuous variables between these two spaces using distance-based Gaussian kernel weighting.

Cluster Annotation Projection: Given cell cluster annotations $c=\{c_{1},c_{2},\ldots ,c_{m}\}$, where $c_{i}$ represents the cluster label for cell *i*, we project these discrete labels onto grid points using nearest neighbor assignment:(22)$$\mathrm{cluster}\left(g\right)=c_{j^{*}}\;\mathrm{where}\;j^{*}=\arg\min_{j\in \{1,\ldots ,m\}}\left|\right|G_{g}-X_{j}{\left|\right|}_{2}$$

The cluster assignment for each mapped grid point, $g\in G_{\mathrm{mapped}}$, is determined by the cluster label of its closest cell in the embedding space.

Grid-to-Cell Projection: For projecting continuous values from grid space back to cells, we employ a Gaussian kernel-weighted interpolation scheme. For each cell *i*, we identify the *k* nearest grid points:(23)$$N_{k}\left(i\right)=\left\{g_{1},g_{2},\ldots ,g_{k}\right\}\;\mathrm{where}\;||X_{i}-G_{g_{j}}{\left|\right|}_{2}\leq \left|\right|X_{i}-G_{g_{j+1}}{\left|\right|}_{2}$$

The local distance variance is computed as:(24)$$\sigma_{i}^{2}=\mathrm{Var}(\{||X_{i}-G_{g}{\left|\right|}_{2}:g\in N_{k}\left(i\right)\})$$

Gaussian weights are calculated for each neighboring grid point:(25)$$w_{i,g}=\frac{\exp\left(-\frac{\left|\right|X_{i}-G_{g}{\left|\right|}_{2}^{2}}{2\sigma_{i}^{2}}\right)}{\sum_{g^{\prime }\in N_{k}\left(i\right)}\exp\left(-\frac{\left|\right|X_{i}-G_{g^{\prime }}{\left|\right|}_{2}^{2}}{2\sigma_{i}^{2}}\right)}$$

The projected value at cell *i* is computed as:(26)$$v_{i}=\sum_{g\in N_{k}\left(i\right)}w_{i,g}\cdot V\left(g\right)$$
where V(g) represents the grid value at position *g*. For optional annotation weighting with factor $w_{\mathrm{ann}}$:(27)$$v_{i}^{\mathrm{weighted}}=v_{i}\cdot \log\left(w_{\mathrm{ann},i}\right)$$

Cell-to-Grid Projection: For projecting cellular data onto grid points, we reverse the projection direction. For each grid point, *g*, we identify the *k* nearest cells:(28)$$M_{k}\left(g\right)=\left\{i_{1},i_{2},\ldots ,i_{k}\right\}\;\mathrm{where}\;||G_{g}-X_{i_{j}}{\left|\right|}_{2}\leq \left|\right|G_{g}-X_{i_{j+1}}{\left|\right|}_{2}$$

The local variance for grid point *g* is:(29)$$\sigma_{g}^{2}=\mathrm{Var}(\{||G_{g}-X_{i}{\left|\right|}_{2}:i\in M_{k}\left(g\right)\})$$

Gaussian weights for neighboring cells are:(30)$$w_{g,i}=\frac{\exp\left(-\frac{\left|\right|G_{g}-X_{i}{\left|\right|}_{2}^{2}}{2\sigma_{g}^{2}}\right)}{\sum_{i^{\prime }\in M_{k}\left(g\right)}\exp\left(-\frac{\left|\right|G_{g}-X_{i^{\prime }}{\left|\right|}_{2}^{2}}{2\sigma_{g}^{2}}\right)}$$

The projected data value at grid point *g* for feature *f* is:(31)$$D_{g,f}=\sum_{i\in M_{k}\left(g\right)}w_{g,i}\cdot D_{i,f}$$
where $D_{i,f}$ represents the data value for feature *f* in cell *i*.

Normalization: For non-negative projections, final values are normalized to the unit interval:(32)$${\tilde{v}}_{i}=\frac{v_{i}-\min_{j}v_{j}}{\max_{j}v_{j}-\min_{j}v_{j}}$$

### 2.5. Reinforcement Learning Environment Configuration

To model cellular fate decisions as a sequential decision-making process, we designed a reinforcement learning environment that transforms the grid representation into a MDP (Markov Decision Process), where agents learn optimal trajectories through reward signals derived from biological annotations.

State Space Construction: For each grid point $g\in G_{\mathrm{mapped}}$, we construct a state representation using the *k*-nearest neighbor cells in the embedding space. Given latent features $Z\in R^{m\times d}$ from principal component analysis:(33)$$N_{k}\left(g\right)=\left\{i_{1},i_{2},\ldots ,i_{k}\right\}\;\mathrm{where}\;||G_{g}-X_{i_{j}}{\left|\right|}_{2}\leq \left|\right|G_{g}-X_{i_{j+1}}{\left|\right|}_{2}$$

The state vector for grid point *g* is computed as:(34)$$s_{g}=\frac{1}{k}\sum_{i\in N_{k}\left(g\right)}Z_{i}$$

Action Space Definition: The action space consists of 8 discrete directional movements corresponding to the 8-connectivity neighborhood:(35)A={R,RT,T,LT,L,LB,B,RB}

For grid position (i,j), the action mapping is defined as:(36)$$\phi \left(a\right)=\left\{\begin{array}{cc} \left(i,j+1\right) & \mathrm{if}\;a=R\\ \left(i+1,j+1\right) & \mathrm{if}\;a=\mathrm{RT}\\ \left(i+1,j\right) & \mathrm{if}\;a=T\\ \vdots & \\ \left(i-1,j+1\right) & \mathrm{if}\;a=\mathrm{RB} \end{array}\right.$$

Discrete Reward Function: For lineage-specific trajectory learning, we define rewards based on cluster annotations and pseudotime progression. Given starting clusters $C_{\mathrm{start}}$ and target clusters $C_{\mathrm{end}}$:(37)$$R_{d}\left(s,a\right)=\left\{\begin{array}{cc} -1 & \mathrm{if}\;\mathrm{transition}\;\mathrm{leads}\;\mathrm{to}\;\mathrm{masked}\;\mathrm{grid}\\ \exp(-\beta \cdot {\hat{t}}_{s^{\prime }})+\Delta t & \mathrm{if}\;s^{\prime }\in C_{\mathrm{end}}\;(\mathrm{Decision}\;\mathrm{mode})\\ 1-\exp(-\beta \cdot {\hat{t}}_{s^{\prime }})+\Delta t & \mathrm{if}\;s^{\prime }\in C_{\mathrm{end}}\;(\mathrm{Contribution}\;\mathrm{mode})\\ 0 & \mathrm{otherwise} \end{array}\right.$$
where ${\hat{t}}_{s^{\prime }}$ is the normalized pseudotime at next state $s^{\prime }$, β is the decay coefficient and $\Delta t=\psi \left(s^{\prime }\right)-\psi \left(s\right)$ represents pseudotime progression.

Continuous Reward Function: For gene expression-guided learning, rewards are computed using projected continuous values:(38)$$\bar{r}\left(g\right)=\frac{1}{|K_{\mathrm{reward}}|}\sum_{k\in K_{\mathrm{reward}}}D_{g,k}$$
where $K_{\mathrm{reward}}$ represents reward gene sets and $D_{g,k}$ is the projected expression of gene *k* at grid *g*. The continuous reward function is:(39)$$R_{c}\left(s,a\right)=\left\{\begin{array}{cc} -1 & \mathrm{if}\;\mathrm{transition}\;\mathrm{leads}\;\mathrm{to}\;\mathrm{masked}\;\mathrm{grid}\\ w_{\mathrm{reward}}\cdot \bar{r}\left(s^{\prime }\right)-w_{\mathrm{punish}}\cdot \bar{p}\left(s^{\prime }\right) & \mathrm{if}\;s^{\prime }\;\mathrm{is}\;\mathrm{valid}\\ 0 & \mathrm{otherwise} \end{array}\right.$$
where the weighting factors are defined as:(40)$$w_{\mathrm{reward}}=\left\{\begin{array}{cc} \exp(-\beta \cdot {\hat{t}}_{s^{\prime }}) & \mathrm{Decision}\;\mathrm{mode}\\ 1-\exp(-\beta \cdot {\hat{t}}_{s^{\prime }}) & \mathrm{Contribution}\;\mathrm{mode} \end{array}\right.$$
and $\bar{p}\left(s^{\prime }\right)$ represents the mean expression of punishment genes.

Episode Termination: Episodes terminate under the following conditions:(41)$$\mathrm{terminate}=\left\{\begin{array}{cc} \mathrm{True} & \mathrm{if}\;s^{\prime }\in G_{\mathrm{boundary}}\setminus G_{\mathrm{start}}\\ \mathrm{True} & \mathrm{if}\;s^{\prime }\in \mathrm{trajectory}\;\mathrm{history}\\ \mathrm{True} & \mathrm{if}\;R(s,a)=-1\\ \mathrm{False} & \mathrm{otherwise} \end{array}\right.$$
with trajectory truncation after $T_{\max}$ steps.

Experience Replay: Training experiences are stored in a circular buffer, B, with capacity $N_{\mathrm{buffer}}$:(42)$$B={\left\{\left(s_{t},a_{t},r_{t},s_{t+1},d_{t}\right)\right\}}_{t=1}^{\left|B\right|}$$
where $d_{t}\in \left\{0,1\right\}$ indicates episode termination. During training, mini-batches of size *B* are uniformly sampled from B for gradient updates.

### 2.6. Reinforcement Learning Model Architecture and Training

To learn optimal cellular fate decision policies, we implemented three distinct reinforcement learning algorithms that leverage different approximation strategies and learning paradigms for trajectory optimization in the grid-based cellular environment.

Tabular Q-Learning: For discrete state spaces, we employ tabular Q-learning with ϵ-greedy exploration. The Q-table is initialized as:(43)$$Q\in R^{|G_{\mathrm{mapped}}|\times 8}$$

The exploration probability decays exponentially with training steps:(44)$$\epsilon \left(t\right)=\epsilon_{\min}+\left(\epsilon_{\max}-\epsilon_{\min}\right)\exp\left(-\frac{0.01\cdot t}{1000}\right)$$
where $\epsilon_{\max}=0.9$, $\epsilon_{\min}=0.01$ and *t* represents the current step. Action selection follows:(45)$$a_{t}=\left\{\begin{array}{cc} \arg\max_{a}Q\left(s_{t},a\right) & \mathrm{with}\;\mathrm{probability}\;1-\epsilon (t)\\ \mathrm{uniform}\;\mathrm{random} & \mathrm{with}\;\mathrm{probability}\;\epsilon (t) \end{array}\right.$$

Q-value updates use temporal difference learning:(46)$$Q\left(s_{t},a_{t}\right)\leftarrow Q\left(s_{t},a_{t}\right)+\alpha \left[r_{t}+\gamma \max_{a^{\prime }}Q\left(s_{t+1},a^{\prime }\right)-Q\left(s_{t},a_{t}\right)\right]$$
where α is the learning rate and γ is the discount factor.

Actor–Critic Architecture: For continuous state representations, we implement an actor–critic framework with separate policy and value networks. The policy network outputs action probabilities:(47)$$\pi_{\theta }\left(a|s\right)=\mathrm{Softmax}\left(1+5\cdot \sigma \left(W_{\pi }^{\left(2\right)}\mathrm{ReLU}\left(W_{\pi }^{\left(1\right)}s+b_{\pi }^{\left(1\right)}\right)+b_{\pi }^{\left(2\right)}\right)\right)$$
where σ denotes the sigmoid activation and the scaling factor ensures exploration. The value network approximates state values:(48)$$V_{\phi }\left(s\right)=W_{v}^{\left(2\right)}\mathrm{ReLU}\left(W_{v}^{\left(1\right)}s+b_{v}^{\left(1\right)}\right)+b_{v}^{\left(2\right)}$$

The temporal difference error is computed as:(49)$$\delta_{t}=r_{t}+\gamma V_{\phi }\left(s_{t+1}\right)\left(1-d_{t}\right)-V_{\phi }\left(s_{t}\right)$$
where $d_{t}\in \left\{0,1\right\}$ indicates episode termination. Policy updates incorporate entropy regularization:(50)$$L_{\pi }=-E\left[\log\pi_{\theta }\left(a_{t}|s_{t}\right)\cdot \delta_{t}\right]+\lambda_{\mathrm{ent}}E\left[\sum_{a}\pi_{\theta }\left(a|s_{t}\right)\log\pi_{\theta }\left(a|s_{t}\right)\right]$$
where $\lambda_{\mathrm{ent}}=0.01$ promotes exploration. The value network loss is:(51)$$L_{v}=E\left[{\left(V_{\phi }\left(s_{t}\right)-\left(r_{t}+\gamma V_{\phi }\left(s_{t+1}\right)\left(1-d_{t}\right)\right)\right)}^{2}\right]$$

DDQN (Double Deep Q-Network): To address overestimation bias in deep Q-learning, we employ a double DQN architecture with target network stabilization. The policy network estimates action values:(52)$$Q_{\theta }\left(s,a\right)=W_{q}^{\left(2\right)}\mathrm{ReLU}\left(W_{q}^{\left(1\right)}s+b_{q}^{\left(1\right)}\right)+b_{q}^{\left(2\right)}$$

Action selection uses decaying ϵ-greedy with:(53)$$\epsilon \left(t\right)=\epsilon_{\mathrm{end}}+\left(\epsilon_{\mathrm{start}}-\epsilon_{\mathrm{end}}\right)\exp\left(-\frac{0.1\cdot t}{\tau_{\mathrm{decay}}}\right)$$

The target value computation follows the double Q-learning update:(54)$$\begin{array}{cc} a^{*} & =\arg\max_{a}Q_{\theta }\left(s_{t+1},a\right) \end{array}$$(55)$$\begin{array}{cc} y_{t} & =r_{t}+\gamma Q_{\theta^{-}}\left(s_{t+1},a^{*}\right)\left(1-d_{t}\right) \end{array}$$
where $Q_{\theta^{-}}$ denotes the target network. The loss function employs smooth L1 loss:(56)$$L_{\mathrm{DQN}}=E\left[\mathrm{SmoothL}1\left(Q_{\theta }\left(s_{t},a_{t}\right)-y_{t}\right)\right]$$

Target network updates use either soft updates:  (57)$$\theta^{-}\leftarrow \left(1-\tau \right)\theta^{-}+\tau \theta$$
or hard updates every *C* steps: $\theta^{-}\leftarrow \theta$.

Training Dynamics: All algorithms track performance through exponential moving averages. For value-based methods, we monitor the maximum Q-value:(58)$${\bar{Q}}_{\max}\left(t\right)=0.005\cdot \max_{a}Q\left(s_{t},a\right)+0.995\cdot {\bar{Q}}_{\max}\left(t-1\right)$$

For actor–critic, we track the state value:(59)$$\bar{V}\left(t\right)=0.005\cdot V_{\phi }\left(s_{t}\right)+0.995\cdot \bar{V}\left(t-1\right)$$

Episode returns are accumulated and averaged over sliding windows of 100 episodes to assess learning progress and convergence.

Experience Replay: For off-policy DDQN training, experiences are stored in a circular buffer with capacity $N_{\mathrm{buffer}}$ and sampled uniformly for mini-batch updates once the buffer contains at least $0.1\cdot N_{\mathrm{buffer}}$ transitions.

### 2.7. scRL Parameter Configuration and Data Structure Organization

The scRL framework employs a hierarchical parameter configuration system and structured data organization to ensure computational efficiency and biological interpretability across diverse single-cell analysis scenarios.

Core Data Container Architecture: scRL organizes computational results through a structured container class with five primary data categories:(60)$$D_{\mathrm{scRL}}=\left\{E,G,Q,S,T\right\}$$
where E represents embedding data, G contains grid information, Q stores reinforcement learning components, S holds simulation data and T maintains trajectory records.

Grid Generation Parameters: The fundamental grid configuration employs two primary parameters. Grid resolution parameter *n* determines spatial discretization:(61)$$|G_{\mathrm{total}}|=n^{2}$$

Observer parameter *j* controls boundary detection sensitivity:(62)$$S_{\mathrm{observer}}=j\times \left|B_{\mathrm{spine}}\right|$$
where $B_{\mathrm{spine}}$ represents the boundary spine points. Default values are configured as $n_{\mathrm{default}}=50$ and $j_{\mathrm{default}}=3$.

Parallel Processing Configuration: Computational parallelization is controlled by the parameter:(63)$$n_{\mathrm{jobs}}\in \left\{1,2,\ldots ,N_{\mathrm{CPU}}\right\}$$
where $N_{\mathrm{CPU}}$ represents available processor cores. The default setting is $n_{\mathrm{jobs}}=8$.

Cluster Projection Parameters: Cell type annotation projection employs cluster assignment vectors:(64)$$C=\left\{c_{1},c_{2},\ldots ,c_{m}\right\}\;\mathrm{where}\;c_{i}\in N$$

Color mapping utilizes predefined palettes or automatic generation:(65)$$P_{\mathrm{color}}=\left\{p_{1},p_{2},\ldots ,p_{\left|C\right|}\right\}\;\mathrm{where}\;p_{i}\in {\left[0,1\right]}^{3}$$

Pseudotime Alignment Parameters: Temporal ordering configuration includes sampling parameters:(66)$$n_{\mathrm{sample}}\in \left\{1,2,\ldots ,\right|C_{\mathrm{early}}\left|\right\}$$

Boundary restriction flag:(67)$$b_{\mathrm{boundary}}\in \left\{0,1\right\}$$

Key identifier for storage:(68)$$k_{\mathrm{add}}\in \left\{\mathrm{strings}\right\}$$
where the default configuration sets $n_{\mathrm{sample}}=10$, $b_{\mathrm{boundary}}=1$ and $k_{\mathrm{add}}=\text{‘}\mathrm{pseudotime}\text{’}$.

Projection Parameters: Bidirectional projection between cells and grids employs neighborhood parameters:(69)$$k_{\mathrm{neighbors}}\in \left\{1,2,\ldots ,\right|G_{\mathrm{mapped}}\left|\right\}$$

Weighting factors for optional annotation scaling:(70)$$w_{\mathrm{annotation}}\in R^{+}$$

Normalization flags:(71)$$f_{\mathrm{negative}}\in \left\{0,1\right\}$$

Default settings utilize $k_{\mathrm{neighbors}}=15$.

Data Structure Hierarchy: The framework maintains structured data organization:(72)$$\begin{array}{cc} E & =\{X_{\mathrm{embedding}},C_{\mathrm{clusters}},P_{\mathrm{colors}},D_{\mathrm{data}}\} \end{array}$$(73)$$\begin{array}{cc} G & =\{n,G_{\mathrm{grids}},M_{\mathrm{masked}},M_{\mathrm{mapped}},A_{\mathrm{adjacency}}\} \end{array}$$(74)$$\begin{array}{cc} Q & =\{R_{\mathrm{rewards}},M_{\mathrm{matrix}},k_{\mathrm{reward}}\} \end{array}$$

### 2.8. Embedding Techniques and Evaluation Framework

To transform high-dimensional single-cell data into interpretable representations and assess the quality of our grid-based trajectory inference, we employ multiple dimensionality reduction techniques and comprehensive evaluation metrics that capture both biological relevance and computational performance.

Principal Component Analysis: Given an n×m gene expression matrix, X, with *n* cells and *m* genes, PCA identifies orthogonal axes that maximize variance:(75)Z=XW
where $W\in R^{m\times k}$ contains the principal component weights and $Z\in R^{n\times k}$ represents cell embeddings in the reduced space. The columns of W are eigenvectors of the covariance matrix $X^{T}X$ corresponding to the *k* largest eigenvalues $\lambda_{1}\geq \lambda_{2}\geq \cdots \geq \lambda_{k}$. The proportion of variance explained by the *i*-th component is:(76)$${\mathrm{var}}_{i}=\frac{\lambda_{i}}{\sum_{j=1}^{m}\lambda_{j}}$$

Nonlinear Embedding Methods: t-SNE constructs probability distributions over cell pairs in high-dimensional and low-dimensional spaces:  (77)$$\begin{array}{cc} p_{ij} & =\frac{p_{j|i}+p_{i|j}}{2n},\;p_{j|i}=\frac{\exp(-||z_{i}-z_{j}|{|}^{2}/2\sigma_{i}^{2})}{\sum_{k\neq i}\exp(-||z_{i}-z_{k}{\left|\right|}^{2}/2\sigma_{i}^{2})} \end{array}$$(78)$$\begin{array}{cc} q_{ij} & =\frac{\left(1+|\right|y_{i}-y_{j}{\left|\right|}^{2}{)}^{-1}}{\sum_{k\neq l}\left(1+|\right|y_{k}-y_{l}{\left|\right|}^{2}{)}^{-1}}\;\;\;\;\;\;\;\;\;\;\;\;\;\;\;\;\;\;\;\;\;\;\;\;\;\; \end{array}$$

The embedding minimizes the Kullback–Leibler divergence: $C\left(Y\right)=\sum_{i\neq j}p_{ij}\log\frac{p_{ij}}{q_{ij}}$.

UMAP constructs a fuzzy topological representation through weighted graphs with connection weights:(79)$$\begin{array}{cc} w_{ij} & =\exp\left(-\frac{d\left(x_{i},x_{j}\right)-\rho_{i}}{\sigma_{i}}\right) \end{array}$$(80)$$\begin{array}{cc} v_{ij} & =\left(1+a\cdot |\right|y_{i}-y_{j}{\left|\right|}^{2b}{)}^{-1} \end{array}$$

The optimization objective combines attraction and repulsion terms:(81)$$\sum_{i,j}w_{ij}\log\frac{w_{ij}}{v_{ij}}+\left(1-w_{ij}\right)\log\frac{1-w_{ij}}{1-v_{ij}}$$

Diffusion Maps: Starting with similarity matrix W, where $W_{ij}=\exp(-||x_{i}-x_{j}{\left|\right|}^{2}/\epsilon )$, we compute the normalized transition matrix:(82)$$\tilde{P}=D^{-1/2}WD^{-1/2}$$
where D is diagonal with $D_{ii}=\sum_{j}W_{ij}$. Diffusion coordinates are defined by eigenvectors $\phi_{k}$ of $\tilde{P}$:(83)$$\Phi_{t}\left(x_{i}\right)=\left(\lambda_{1}^{t}\phi_{1}\left(i\right),\lambda_{2}^{t}\phi_{2}\left(i\right),\ldots ,\lambda_{k}^{t}\phi_{k}\left(i\right)\right)$$

scVI (single-cell Variational Inference): scVI employs variational autoencoders to model single-cell gene expression through a hierarchical generative process. The latent representation $z_{n}∼N\left(0,I\right)$ captures cellular states, while library size factors $l_{n}∼\mathrm{LogNormal}\left(\mu_{l},\sigma_{l}^{2}\right)$ account for technical variation. Gene expression rates are modeled through neural networks:(84)$$\rho_{ng}=f_{h}^{\left(1\right)}{\left(z_{n},s_{n}\right)}_{g}\;\mathrm{and}\;\alpha_{ng}=f_{h}^{\left(2\right)}{\left(z_{n},s_{n}\right)}_{g}$$

The observation model follows a zero-inflated negative binomial distribution:(85)$$x_{ng}∼\mathrm{ZINB}\left(\mu_{ng},\theta_{g},\pi_{ng}\right)$$
where $\mu_{ng}=l_{n}\rho_{ng}$ and zero-inflation probability $\pi_{ng}=f_{h}^{\left(3\right)}{\left(z_{n},s_{n}\right)}_{g}$. The variational approximation $q_{\phi }\left(z_{n}|x_{n},s_{n}\right)=N\left(\mu_{n}^{\phi },\sigma_{n}^{\phi }\right)$ optimizes the evidence lower bound:(86)$$L\left(\phi ,h\right)=E_{q_{\phi }}\left[\log p_{h}\left(x_{n}|z_{n},s_{n}\right)\right]-\mathrm{KL}\left[q_{\phi }\left(z_{n}|x_{n},s_{n}\right)∥p\left(z_{n}\right)\right]$$

Linear-scVI Extensions: Linear-scVI enhances interpretability by imposing linear structure on the latent space:(87)$$z_{n}=Ac_{n}+\epsilon_{n}$$
where $c_{n}∼N\left(0,I_{k}\right)$ represents interpretable linear factors and $A\in R^{d\times k}$ provides the transformation matrix. Cellular trajectories are parameterized through time-dependent dynamics:(88)$$c_{n}\left(t\right)=Bt_{n}+\nu_{n}$$
where B captures trajectory directions. The decoder employs linear combinations:(89)$$\log\rho_{ng}=W_{g}^{T}z_{n}+b_{g}+\log l_{n}$$

The modified ELBO incorporates linear constraints:(90)$$L_{\mathrm{linear}}=E_{q_{\phi }}\left[\log p\left(x_{n}|z_{n}\right)\right]-\mathrm{KL}\left[q_{\phi }\left(c_{n}|x_{n}\right)∥p\left(c_{n}\right)\right]-\mathrm{KL}\left[q_{\phi }\left(z_{n}|c_{n},x_{n}\right)∥p\left(z_{n}|c_{n}\right)\right]$$

Latent Dirichlet Allocation: LDA identifies latent gene expression programs through topic modeling. The generative process defines gene expression programs $\beta_{k}∼\mathrm{Dirichlet}\left(\eta \right)$ and cell-specific topic proportions $\theta_{n}∼\mathrm{Dirichlet}\left(\alpha \right)$. Gene assignments follow $z_{ng}∼\mathrm{Categorical}\left(\theta_{n}\right)$ and expression levels $w_{ng}∼\mathrm{Categorical}\left(\beta_{z_{ng}}\right)$. Variational inference employs coordinate ascent updates:(91)$$\begin{array}{cc} \phi_{ngk} & \propto \exp\left\{\Psi \left(\gamma_{nk}\right)+\Psi \left(\lambda_{kw_{ng}}\right)-\Psi \left(\sum_{v}\lambda_{kv}\right)\right\} \end{array}$$(92)$$\begin{array}{cc} \gamma_{nk} & =\alpha_{k}+\sum_{g=1}^{G_{n}}\phi_{ngk}\;\;\;\;\;\;\;\;\;\;\;\;\;\;\;\;\;\;\;\;\;\;\;\;\;\;\;\;\;\;\;\; \end{array}$$(93)$$\begin{array}{cc} \lambda_{kv} & =\eta_{v}+\sum_{n=1}^{N}\sum_{g:w_{ng}=v}\phi_{ngk}\;\;\;\;\;\;\;\;\;\;\;\;\;\;\;\;\;\;\;\;\;\;\;\; \end{array}$$

External Evaluation Metrics: ARI corrects for random clustering effects:(94)ARI=∑ijnij2−∑iai2∑jbj2/n212∑iai2+∑jbj2−∑iai2∑jbj2/n2
where $n_{ij}$ represents cells belonging to both true class *i* and predicted class *j*. NMI quantifies information-theoretic consistency:(95)$$\mathrm{NMI}\left(U,V\right)=\frac{2\times I(U,V)}{H(U)+H(V)}$$

Internal Evaluation Metrics: ASW measures within-cluster cohesion versus between-cluster separation:(96)$$s\left(i\right)=\frac{b(i)-a(i)}{\max\{a(i),b(i)\}},\;\mathrm{ASW}=\frac{1}{n}\sum_{i=1}^{n}s\left(i\right)$$

CH evaluates the ratio of between-cluster to within-cluster dispersion:(97)$$\mathrm{CH}=\frac{B(k)/(k-1)}{W(k)/(n-k)}$$

DB assesses intra-cluster scatter relative to inter-cluster distances:(98)$$\mathrm{DB}=\frac{1}{k}\sum_{i=1}^{k}\max_{j\neq i}\left(\frac{S_{i}+S_{j}}{d_{ij}}\right)$$

Comprehensive Scoring: To enable fair comparison across metrics with different scales and optimization directions, we apply min–max normalization:(99)$$X_{\mathrm{scaled}}=\frac{X-X_{\min}}{X_{\max}-X_{\min}}$$

For metrics requiring minimization (DB), we use $X_{\mathrm{scaled}}=\frac{X_{\max}-X}{X_{\max}-X_{\min}}$.

We define two key metrics to quantitatively assess the alignment of cell lineage intensity with learned dimensionality reduction embeddings.

MAS (Manifold Alignment Score): Given a lineage intensity vector, *L*, and the two-dimensional UMAP manifold represented by its orthogonal dimensions $U_{1}$ and $U_{2}$, the MAS is defined as the average of the absolute Pearson correlation coefficients between the lineage intensity vector and each UMAP dimension:(100)$$\mathrm{MAS}=\frac{1}{2}\left(|\rho \left(L,U_{1}\right)|+|\rho \left(L,U_{2}\right)|\right)$$
where ρ(X,Y) denotes the Pearson correlation coefficient between variables *X* and *Y*.

PAAC (Principal Axis Alignment Coefficient): Given a lineage intensity vector, *L*, and the first two principal components obtained from PCA, denoted as $P_{1}$ and $P_{2}$, the PAAC is defined as the average of the absolute Pearson correlation coefficients between the lineage intensity vector and these two principal components:(101)$$\mathrm{PAAC}=\frac{1}{2}\left(|\rho \left(L,P_{1}\right)|+|\rho \left(L,P_{2}\right)|\right)$$

Similarly, ρ(X,Y) denotes the Pearson correlation coefficient.

### 2.9. Comprehensive Validation Framework for scRL

To establish scRL’s robustness across diverse biological contexts, we designed a systematic validation strategy encompassing five complementary analytical dimensions, each addressing critical aspects of cellular fate decision modeling through carefully selected single-cell datasets.

Benchmark Performance Validation: We assessed scRL’s fundamental capabilities using human bone marrow ${\mathrm{CD34}}^{+}$ hematopoietic progenitor cells (5780 cells, S-SUBS8) and mouse endocrinogenesis datasets (2531 cells, GSE132188). These canonical differentiation systems enabled direct comparison with 8 existing methodologies across 5 unsupervised metrics, focusing on pseudotime ordering accuracy, lineage contribution strength quantification and fate decision intensity detection. For the hematopoietic dataset, we configured scRL with grid parameters (n=50, j=3), defining erythroid (Ery_1 [${\mathrm{Erythrocyte}}_{1}$], Ery_2 [${\mathrm{Erythrocyte}}_{2}$], Mega [Megakaryocyte]), myeloid (Mono_1 [${\mathrm{Monocyte}}_{1}$], Mono_2 [${\mathrm{Monocyte}}_{2}$], DCs [dendritic cells]), and lymphoid (CLP [Common Lymphoid Progenitor]) lineages as reward targets from HSC_1 (Hematopoietic Stem ${\mathrm{Cell}}_{1}$) starting points. The endocrinogenesis analysis targeted alpha, beta, delta, and epsilon cell fates from *Ngn3* low/high and *Fev*^+^ progenitors, enabling systematic performance benchmarking across distinct developmental contexts.

Temporal Precedence Analysis: Using human phenotypical Hematopoietic Stem Cell datasets (GSE117498) containing HSC (Hematopoietic Stem Cell), MPP (Multipotent Progenitor), MLP (Multipotent Lymphoid Progenitor), CMP (Common Myeloid Progenitor), MEP (Megakaryocyte–Erythroid Progenitor), GMP (Granulocyte–Monocyte Progenitor), and PreBNK (Pre-B/NK Progenitor) populations, we validated whether scRL-identified fate decision states represent early specification events preceding observable lineage contributions. This analysis specifically examined the temporal relationship between computational fate predictions and experimentally characterized progenitor hierarchies. By targeting MEP and GMP fates from HSC origins, we established that scRL captures pre-contribution cellular states that systematically anticipate downstream differentiation events, confirming the framework’s predictive temporal resolution.

Multi-scale Decision Integration: The AML (acute myeloid leukemia) dataset (94,311 cells, GSE185993) provided a pathological context for validating coherence between molecular-level and cellular-level decision processes. This large-scale analysis demonstrated that fate decision intensity and gene decision intensity characterize equivalent cellular states, confirming scRL’s capacity to integrate multi-scale biological information. The dataset’s cellular heterogeneity and malignant transformation context enabled assessment of scRL’s robustness under conditions where normal developmental hierarchies are disrupted.

Perturbation Response Validation: We employed conditional *Dapp1* knockout Hematopoietic Stem Cell datasets (10,224 cells, GSE277292) to validate the functional significance of scRL-identified dynamical genes through genetic perturbation analysis. Quality control criteria included total counts (1000–70,000), mitochondrial genes (<5%), ribosomal genes (10–50%) and expressed genes (>1000), followed by Harmony integration and Leiden clustering (resolution 0.6). This perturbation study confirmed that genetic alterations systematically modify fate decision landscapes, establishing biological relevance of scRL-captured decision intensities and validating the framework’s sensitivity to functional genetic modifications.

Pathological Condition Analysis: Radiation-induced hematopoietic injury datasets (41,252 cells, GSE278673) enabled validation of scRL’s performance under stress conditions that alter normal differentiation dynamics. Stringent quality control (total counts 5000–70,000, mitochondrial genes <5%, ribosomal genes 10–50%, expressed genes 1500–7000) preceded Harmony integration and UMAP analysis (min_dist = 1.5). This pathological validation demonstrated scRL’s capacity to capture altered fate decision dynamics, differentiation biases and recovery processes, establishing the framework’s utility for analyzing cellular responses to environmental perturbations and pathological states.

## 3. Results

### 3.1. LDA Provides an Interpretable Latent Space for Single-Cell Lineage Analysis

We conducted a comprehensive comparison of dimensionality reduction methods including deep learning-based approaches (scVI, Linear scVI, and LDA) and traditional methods (PCA, ICA, FA, NMF, and Diff [Diffusion Map]) to identify techniques that effectively capture lineage relationships for investigating cell fate decision mechanisms. Using hematopoietic and pancreatic single-cell datasets, we systematically evaluated these approaches across varying numbers of highly variable genes (1000–5000) and latent dimensions (5–25) using five discriminative metrics: ARI, NMI, ASW, CH, and DB. LDA achieved the highest overall performance scores across all evaluation metrics (Figure 2A,B), with substantial improvements over baseline methods including scVI (+0.785 across HVG [highly variable gene] sizes, +0.713 across latent dimensions) and ICA (+0.675 and +0.618, respectively) (Table 1 and Table 2). Individual latent component analysis showed that LDA exhibited strong subgroup specificity in intensity distributions within original cell type annotations (Figure 2C). Parallel analyses on the endocrinogenesis dataset confirmed these results, with LDA showing enhanced performance gains over scVI (+0.806 across HVG sizes, +0.839 across latent dimensions) and ICA (+0.789 and +0.765, respectively) (Figure A2, Table A1 and Table A2).
Figure 2LDA exhibits superior interpretability in comparative analysis of dimensionality reduction methods. (S-SUBS8) (**A**,**B**) Comparison of interpretability metrics including ARI, NMI, ASW, CH and DB for dimensionality reduction methods (scVI, ICA, PCA, LscVI, FA, Diff, NMF and LDA) evaluated across varying numbers of highly variable genes (1000, 2000, 3000, 4000, 5000) and latent space components (5, 10, 15, 20, 25). (**C**) Intensity distributions of latent components obtained by LDA across different subgroups, demonstrating subgroup specificity.
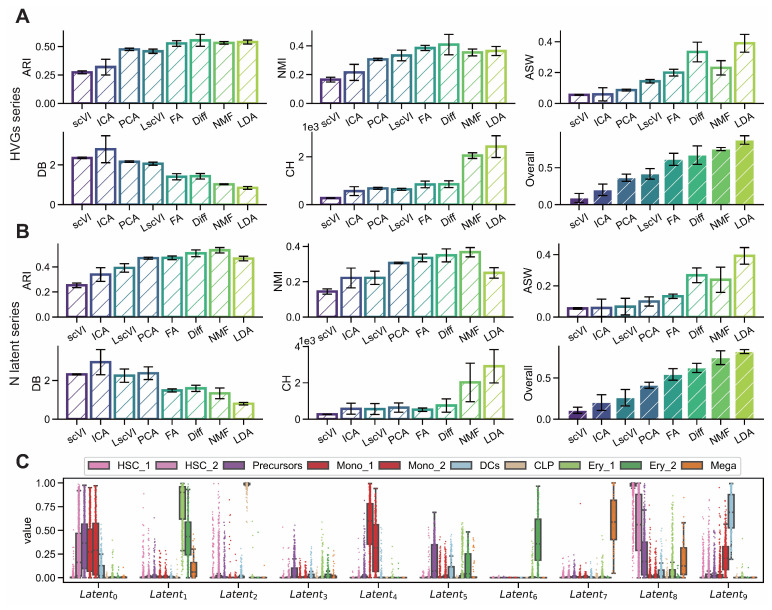



### 3.2. Reinforcement Learning Captures Early Fate-Decision Signals

scRL constructs a grid embedding framework through edge detection-based algorithms, employing reinforcement learning to address the sequential decision-making problem inherent in cellular differentiation and quantify fate decision intensity at any developmental stage (Figure 3A). We utilized intensities from each LDA latent space component as lineage markers and projected them onto two-dimensional UMAP embeddings, revealing clear correspondence between these lineage intensities and distinct differentiation branches across both hematopoietic and pancreatic endocrinogenesis datasets (Figure 3B and Figure A3A). Critical to scRL’s innovation is its ability to identify pre-commitment cellular states: when projecting scRL-derived fate decision intensity onto developmental trajectories, peak decision regions consistently preceded observable lineage specification events, indicating that scRL captures regulatory checkpoints before overt lineage commitment becomes apparent. To quantitatively validate scRL’s superior interpretability, we compared its performance against percentile-based measures of LDA lineage intensities (95%, 90%, 85%, and 80% percentiles) using unsupervised metrics including ASW, CH, and DB (Figure 3C and Figure A3B). Quantitative analysis demonstrated scRL’s consistent superiority across both datasets, with substantial improvements of +0.986, +0.932, +0.735, and +0.375 over LDA at 80%, 85%, 90% and 95% percentiles, respectively, on the hematopoietic dataset (Table 3), and comparable improvements of +0.848, +0.549, +0.371 and +0.287 over corresponding LDA percentile thresholds on the endocrinogenesis dataset (Figure A3, Table A3).
Figure 3scRL demonstrates superior interpretability in fate decision intensity analysis. (S-SUBS8) (**A**) Workflow illustration of scRL’s grid embedding construction and reinforcement learning framework for fate decision intensity inference. (**B**) Projection of lineage intensities from LDA latent space components onto UMAP embeddings showing correspondence with differentiation branches in hematopoietic and pancreatic datasets. (**C**) Comparison of scRL fate decision intensity with LDA percentile variants (95%, 90%, 85%, 80%) using ASW, CH and DB interpretability metrics on hematopoietic dataset. The overall score includes data from ARI and NMI.
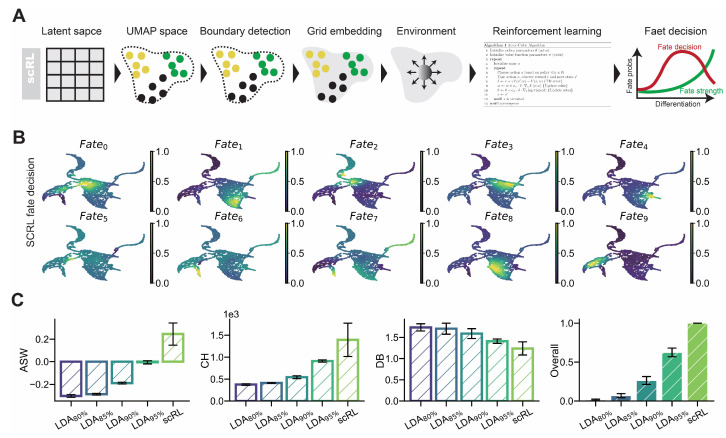



### 3.3. scRL Outperforms Alternative Approaches in Fate-Decision Inference

We evaluated scRL’s fate decision intensity performance against various machine learning and deep learning dimensionality reduction methods including Diff, FA, ICA, LscVI, NMF, PCA and scVI across hematopoietic and endocrinogenesis datasets (Figure 4 and Figure A4A). Using scRL-derived grid embeddings and pseudotime information as the foundation, we extracted 95%, 90%, 85% and 80% percentile intensities from each method as alternative fate decision measures, conducting experiments across HVG sequences from 1000 to 5000 and evaluating performance using unsupervised metrics ARI, NMI, ASW, DB and CH. scRL consistently achieved the highest overall scores across all experimental conditions (Figure 4 and Figure A4B). On the hematopoietic dataset, scRL showed substantial improvements over all baseline methods, with pronounced advantages against ICA-based approaches (overall improvements ranging from +0.614 to +0.811) and scVI-based methods (overall improvements from +0.609 to +0.733) (Table 4). Validation on the endocrinogenesis dataset confirmed these findings, where scRL maintained superior performance with the most significant improvements observed against Diffusion Map approaches (overall improvements from +0.461 to +0.811) and ICA-based methods (overall improvements from +0.612 to +0.812) (Table A4). scRL’s performance advantages were consistently maintained across different percentile thresholds, with higher improvements generally observed at lower percentile values (80–85% percentiles) compared with higher percentiles (95% percentile), indicating that scRL’s reinforcement learning framework effectively captures early fate decision signals that precede observable lineage contribution events.
Figure 4scRL exhibits superior interpretability in fate decision intensity compared with various dimensionality reduction methods. (S-SUBS8) Comprehensive comparison of interpretability metrics for scRL fate decision intensity against Diff, FA, ICA, LscVI, NMF, PCA and scVI at 95%, 90%, 85% and 80% percentile intensities on hematopoietic dataset.
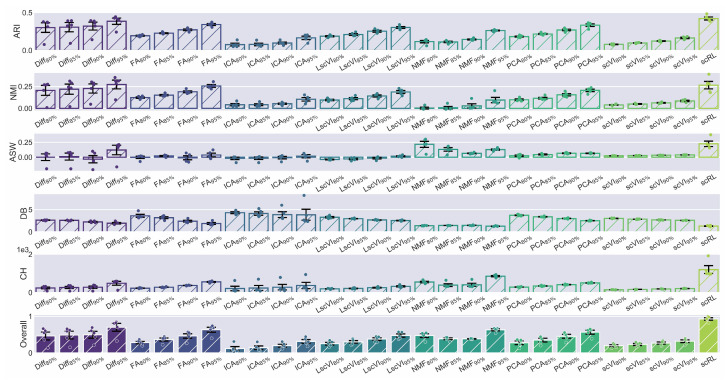



### 3.4. Label-Free scRL Accurately Quantifies Lineage Contribution

When accurate lineage probabilities are unavailable or cannot be mapped to subgroup trajectory branches, scRL offers an alternative solution leveraging reinforcement learning principles with clustering labels as reward signals to evaluate lineage contribution intensity. We applied this approach to hematopoietic and endocrinogenesis datasets using KMeans clustering to obtain labels for distinct subgroups, with scRL computing lineage contribution intensity that exhibited clear subgroup specificity (Figure 5A). We evaluated scRL’s interpretability against established dimensionality reduction techniques (PCA, ICA, FA, NMF, Diff, scVI, LscVI) across varying KMeans subgroup numbers (4, 6, 8, 10, 12) and highly variable gene sequences (1000–5000) using ASW, DB and CH metrics. scRL achieved the best overall assessment scores across all experimental conditions (Figure 5B), with substantial improvements over baseline methods including pronounced advantages against scVI-based approaches (overall improvements from +0.856 to +0.924), ICA methods (+0.409 to +0.694) and PCA approaches (+0.329 to +0.515) across different cluster configurations (Table 5). Validation on the endocrinogenesis dataset confirmed these findings, where scRL maintained superior performance with clear subgroup specificity in both KMeans labels and contribution intensities, showing distinct intensity distribution patterns across lineage branches and consistently strong performance advantages over scVI (overall improvements from +0.856 to +0.924) and ICA (+0.409 to +0.694) across all cluster configurations (Figure A5A,B, Table A5).
Figure 5scRL constructs superior lineage intensity using subgroup information compared with various methods. (S-SUBS8) (**A**) Distribution of 5 subgroups under KMeans clustering and corresponding subgroup lineage intensities obtained by scRL projected onto UMAP embedding, demonstrating subgroup specificity. (**B**) Comprehensive comparison of scRL lineage contribution intensity against PCA, ICA, FA, NMF, Diff, scVI and LscVI using ASW, DB and CH metrics. Evaluations conducted across KMeans subgroup numbers (4, 6, 8, 10, 12) and highly variable gene sequences (1000–5000) as experimental replicates.
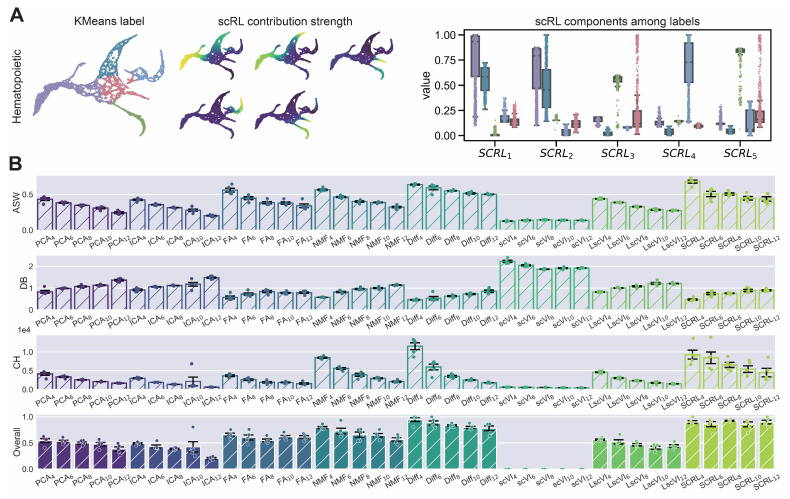



### 3.5. scRL Rivals State-of-the-Art Pseudotime and Fate Inference Tools

We conducted comprehensive performance evaluations comparing scRL with established methods for pseudotime reconstruction (Wishbone [37], Palantir [38], DPT (Diffusion Pseudotime) [39], Monocle2 [40], Monocle3 [12]) and lineage probability inference (Palantir [38], FateID [41], CellRank [42]) using Manifold Alignment Score and Principal Axis Alignment Coefficient metrics. On the hematopoietic dataset, scRL demonstrated superior pseudotime reconstruction performance, achieving a Manifold Alignment Score of 0.773 (+0.091 improvement over the previous best method) and Principal Axis Alignment Coefficient of 0.848 (+0.127 improvement) (Figure 6A,B). For lineage probability assessment, scRL consistently outperformed all baseline methods with a Manifold Alignment Score of 0.323 (+0.085 improvement) and Principal Axis Alignment Coefficient of 0.373 (+0.048 improvement) (Figure 6C,D). Validation on the endocrinogenesis dataset confirmed these findings, where scRL maintained superior performance with pseudotime reconstruction achieving a Manifold Alignment Score of 0.865 (+0.061 improvement over the previous best method) and fate probability analysis demonstrating robust performance with a score of 0.674 (+0.312 improvement over the previous best method) (Figure A6A,B).

### 3.6. scRL Decision Intensities Reveal Pre-Contribution States in Hematopoiesis

We applied scRL to a human hematopoietic progenitor cell dataset containing diverse immune phenotypes [43], conducting pseudotime reconstruction and comprehensive evaluation of lineage contribution strength and fate decision intensity for MEP (Megakaryocyte–Erythroid Progenitor) and GMP (Granulocyte-Monocyte Progenitor) lineages (Figure 7A). scRL revealed distinct temporal dynamics where fate decision intensity and lineage contribution strength exhibited complementary temporal profiles along pseudotime trajectories, with pseudotime-weighted analysis demonstrating that decision-weighted values consistently preceded contribution-weighted values, indicating scRL captures earlier fate specification events before observable lineage contribution (Figure 7B). To validate scRL’s predictive capacity for identifying pre-expression cellular states, we analyzed key transcription factors *GATA1* and *CEBPA*, which serve as master regulators of MEP and GMP differentiation, respectively, with *GATA1* essential for erythroid and megakaryocytic development and *CEBPA* functioning as a master regulator of myeloid lineage contribution. Both transcription factors exhibited peak expression at terminal differentiation stages in their respective lineages, while scRL fate decision analysis demonstrated that both displayed high decision intensity values in more primitive regions compared with their contribution values, suggesting early fate specification events preceding actual gene expression (Figure 7C). Detailed temporal analysis revealed that, for both *GATA1* in MEP lineage and *CEBPA* in GMP lineage, decision state values peaked at intermediate differentiation stages systematically preceding expression peaks, while contribution state values reached maximum intensity at terminal stages and exhibited stronger correlation with pseudotime compared with original expression patterns, with pseudotime-weighted analysis consistently showing decision-weighted values occurred earlier than contribution-weighted values (Figure 7D).

### 3.7. Integrated Gene- and Lineage-Level Decisions Resolve AML Branch Points

We applied scRL’s pseudotime analysis framework to an AML dataset to elucidate the relationship between gene-level and lineage-level decision-making processes during cellular differentiation [44]. Following cellular clustering and pseudotime reconstruction revealing distinct cellular populations and developmental trajectories (Figure 8A,B), we identified four distinct branches representing different lineage decision trajectories (Figure 8C). For each branch, we selected 10 marker genes based on differential expression analysis, requiring expression in at least 25% of cells within the target branch while being expressed in less than 25% of reference cells (Figure 8D), then conducted parallel analyses of gene decision values and lineage decision values across all four branches (Figure 8E,F). Correlation analysis revealed strong relationships between lineage and gene decision values within each branch, demonstrating coordinated decision-making processes at both molecular and cellular levels (Figure 8G), while temporal analysis showed that weighted average pseudotime of lineage decision values consistently preceded actual gene expression across all four branches (Figure 8H). Further characterization through binned pseudotime analysis revealed that lineage decision values across the four branches showed more distinct trajectory dynamics compared with mean Pearson correlation values with branch-specific cells (Figure 8I,J), while gene decision values along binned pseudotime provided clearer temporal dynamics compared with Pearson correlation values with branch-specific genes (Figure 8K,L).
Figure 6Comparative performance analysis of scRL with pseudotime and fate inference methods. (S-SUBS8) (**A**) Pseudotime reconstruction performance comparison using Manifold Alignment Score across methods including Wishbone, Palantir, DPT, Monocle2, Monocle3 and scRL on hematopoietic dataset. (**B**) Principal Axis Alignment Coefficient comparison for pseudotime reconstruction methods. (**C**) Fate probability inference performance using Manifold Alignment Score comparing FateID, Palantir, CellRank and scRL. (**D**) Principal Axis Alignment Coefficient comparison for fate probability inference methods. Green bars indicate performance improvements over previous best methods.
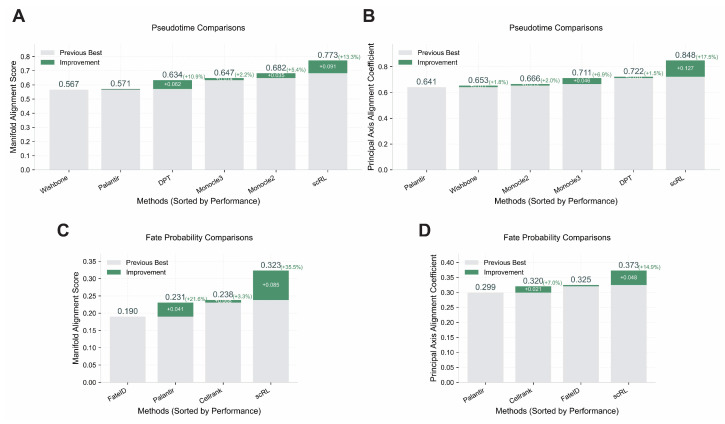

Figure 7scRL application to hematopoietic dataset reveals temporal precedence of decision over contribution states. (GSE117498) (**A**) Standard analysis workflow including pseudotime reconstruction and scRL-derived lineage contribution strength and fate decision intensity results for the two major lineages, MEP and GMP. (**B**) Temporal dynamics of fate decision intensity and lineage contribution strength along pseudotime, with bar plots showing pseudotime-weighted values for both intensities (lower values correspond to more primitive developmental stages). (**C**) Original expression patterns of key transcription factors *GATA1* and *CEBPA* corresponding to MEP and GMP lineages, respectively, alongside scRL-derived contribution strength and decision strength when these genes serve as reward signals. (**D**) Temporal profiles along pseudotime showing original expression intensity, decision strength, contribution strength, and pseudotime-weighted decision and contribution intensities for *GATA1* and *CEBPA* (lower values correspond to more primitive stages), with correlation comparison between contribution strength and original expression relative to pseudotime (higher values indicate superior differentiation trajectory matching).
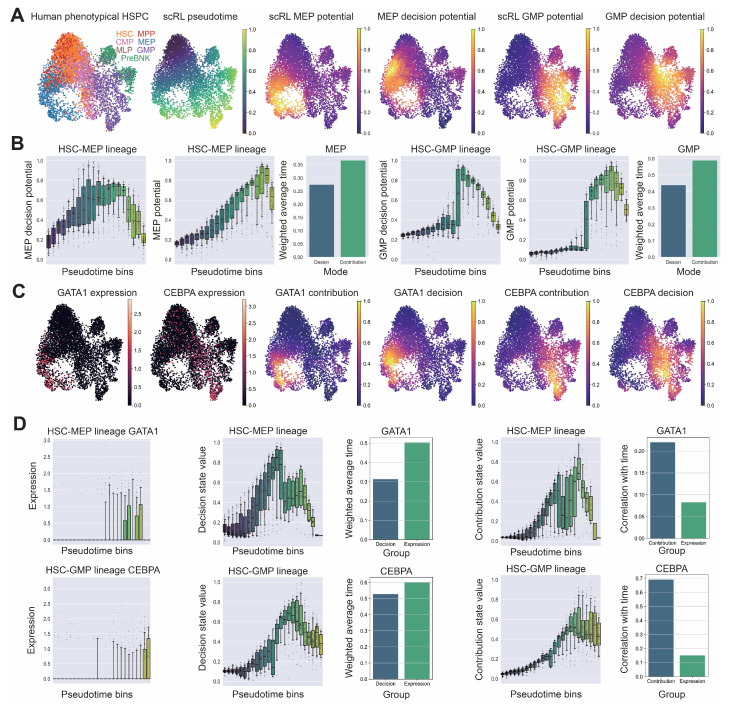

Figure 8Integrative analysis of lineage and gene decision states in acute myeloid leukemia progression. (GSE185993) (**A**) UMAP visualization of AML dataset colored by cellular clusters. Different colors represent distinct cell subpopulations. (**B**) Pseudotime trajectory inferred by scRL across the developmental landscape. (**C**) Four distinct branches identified within the UMAP embedding space. (**D**) Top 10 marker genes for each branch, selected based on log-fold changes with expression criteria of at least 25% cells in target branch and less than 25% in reference cells. (**E**) Gene set decision values computed for each of the four branches. (**F**) Lineage decision values determined for each branch using scRL framework. (**G**) Pearson correlation heatmap between gene set decision values (G1-4) and lineage decision values (L1-4) for each branch. (**H**) Comparison of average pseudotime weighted by lineage decision values versus gene expression for each branch. (**I**) Mean Pearson correlation with branch-specific cells along uniformly 50-binned pseudotime for the complete dataset (top) and trunk region (bottom). (**J**) Lineage decision values for each branch along uniformly 50-binned pseudotime for the complete dataset (top) and trunk region (bottom). (**K**) Mean expression of branch-specific gene sets along uniformly 50-binned pseudotime for the complete dataset (top) and trunk region (bottom). (**L**) Gene decision values of branch-specific gene sets along uniformly 50-binned pseudotime for the complete dataset (top) and trunk region (bottom).
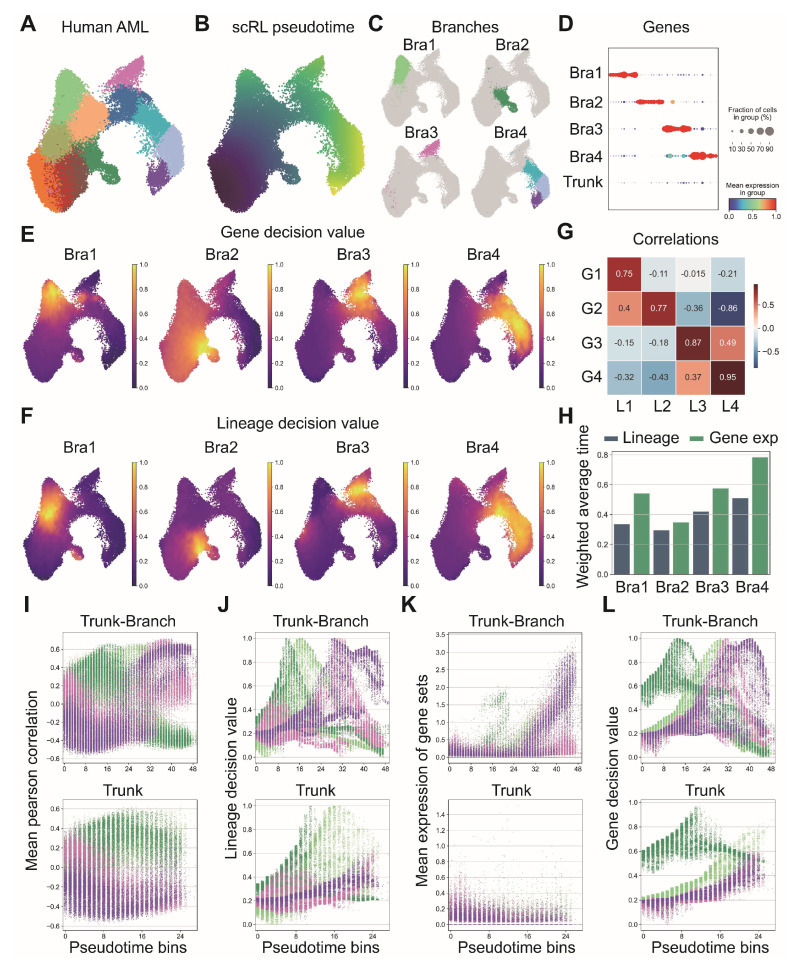



### 3.8. scRL Pinpoints *Dapp1* as an Early Regulator of HSC Fate

We developed a comprehensive workflow for discovering critical dynamical genes in HSC differentiation, establishing a systematic approach beginning with LSK (${\mathrm{Lin}}^{-}{\mathrm{Sca1}}^{+}{\mathrm{ckit}}^{+}$) single-cell data embeddings to identify differentiation starting points, employing velocity analysis for top dynamical genes, utilizing pseudotime analysis for earliest expressing genes, and intersecting highest expressed and most dynamic genes followed by time correlation scoring to identify *Dapp1* as a critical early differentiation regulator (Figure 9A). Comprehensive trajectory analysis using Monocle2 identified 280 genes through intersection of dynamical genes and BEAM (Branching Expression Analysis Modeling) test significant genes with enrichment in hematopoietic differentiation-related GOBP (Gene Ontology Biological Process) terms, while gene correlation analysis using LSK datasets revealed *Dapp1* exhibited strong correlation with myeloid lineage marker *Dach1* and minimal correlation with lymphoid marker *Dntt* (Figure A7A). To validate scRL’s capacity to capture perturbation effects, we constructed a single-cell atlas from bone marrow LSK cells with conditional *Dapp1* knockout, annotating cellular populations based on key transcription factors including *Hlf* and *Tcf15* for primitive HSCs, *Gata1* and *Klf1* for erythroid bias, *Id2* for dendritic cells, *Irf8* for monocytes, *Cebpe* for neutrophils, *Ebf1* for B cells, *Hoxa9* for multipotency, *Satb1* for lymphoid bias and *Gfi1* for myeloid bias (Figure A7B). The *Dapp1* knockout atlas revealed clusters annotated as early clusters (1, 2), multipotent clusters (3, 5, 7), lymphoid clusters (4, 9), myeloid cluster (6) and erythroid clusters (8, 10), with comparative analysis demonstrating significant disruption in lineage contribution homeostasis through altered cluster distributions and pseudotime patterns (Figure 9B,C). Pseudotime analysis and expression patterns of maturation markers *Cd34*, *Mpo* and *Ctsg* indicated impediment to cellular maturation following *Dapp1* deletion (Figure 9D). Analysis along uniformly 50-binned pseudotime revealed *Dapp1* deficiency led to significant perturbations in lineage decision trajectories with decreased decision values for myeloid and erythroid lineages while lymphoid lineage decision values increased (Figure 9E,F). Gene decision values for *Mpo* (myeloid), *Itga2b* (erythroid) and *Dntt* (lymphoid) showed notable alterations along pseudotime, with *Dapp1* deletion decreasing decision values for *Mpo* and *Itga2b* while increasing *Dntt* decision values (Figure 9G,H). Detailed cluster-specific analysis revealed that, within earliest cluster (cluster 1), both myeloid lineage and *Mpo* gene decision trajectories were significantly disrupted with decreased decision values after knockout (Figure 9I,J), while, within multipotent clusters, lymphoid lineage and *Dntt* gene decision values were enhanced following *Dapp1* deletion (Figure 9K). Comprehensive trajectory analysis comparing wild-type and knockout conditions demonstrated distinct alterations in myeloid, erythroid and lymphoid differentiation trajectories with cluster-specific responses to *Dapp1* perturbation, where lymphoid lineage decisions remained largely unchanged in early clusters but were enhanced in multipotent clusters, while erythroid and myeloid decision values showed varying degrees of disruption across different developmental stages (Figure A7C and Figure A8A–E).
Figure 9Lineage contribution of HSCs at the earliest primitive stage perturbed by knockout of critical dynamical gene *Dapp1*. (GSE277292) (**A**) Systematic workflow for identifying critical dynamical gene *Dapp1* in HSC differentiation, beginning with LSK single-cell data embeddings to identify differentiation starting points, employing velocity analysis for top dynamical genes, using pseudotime for earliest expressing genes, intersecting highest expressed and most dynamic genes, and ranking through time correlation scoring. (**B**) *Dapp1* knockout LSK atlas with annotated clusters: 1, 2 as early clusters, 3, 5, 7 as multipotent clusters, 4, 9 as lymphoid clusters, 6 as myeloid cluster and 8,10 as erythroid clusters. (**C**) Cluster distributions and pseudotime comparison between control and knockout groups. (**D**) Box plots comparing pseudotime and expression of maturation markers *Cd34*, *Mpo* and *Ctsg* between conditions. (**E**) Lineage decision values for myeloid, erythroid and lymphoid lineages along uniformly 50-binned pseudotime. (**F**) Box plots comparing lineage decision values across myeloid, erythroid and lymphoid lineages between conditions. (**G**) Gene decision values for *Mpo*, *Itga2b* and *Dntt* along uniformly 50-binned pseudotime. (**H**) Box plots comparing gene decision values for *Mpo*, *Itga2b* and *Dntt* between conditions. (**I**) Myeloid lineage and *Mpo* decision values within early cluster 1 along binned pseudotime. (**J**) Box plots comparing myeloid lineage and *Mpo* decision values within early cluster 1. (**K**) Box plots comparing lymphoid lineage and *Dntt* decision values within multipotent cluster 5.
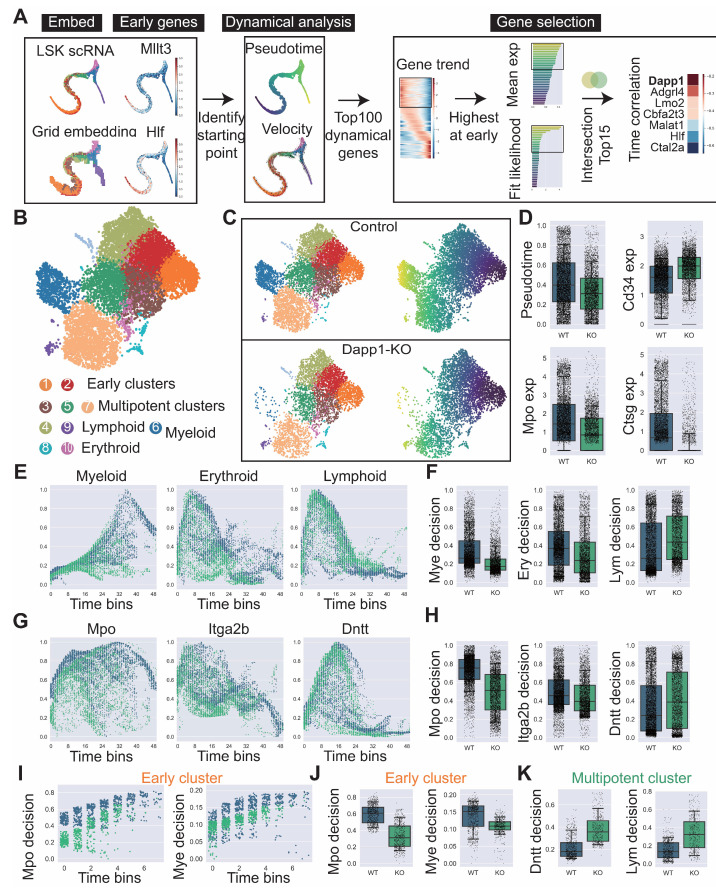



### 3.9. scRL Charts Dynamic Fate Decisions During Post-Irradiation HSC Recovery

We applied scRL to analyze HSC fate decision dynamics under pathological conditions by collecting LSK cells from mice subjected to total body ionizing radiation at multiple time points (days 2, 5, 8, 11, 14, 21, 30) and performing single-cell RNA sequencing to construct a comprehensive temporal atlas. Cellular populations were annotated based on characteristic marker expression patterns including *Kit* and *Ly6a* for LSK identification, *Cd34* and *Flt3* for Multipotent Progenitor components, and lineage-specific markers such as *Il7r* for lymphoid bias, *Cd19* for B cells, *Cd3e* for T cells, and *Itgam* and *Fcer1g* for myeloid bias (Figure A9A). Eleven clusters were identified and annotated as HSC, MPP2/3 (Multipotent Progenitors 2/3), MPP4 (Multipotent Progenitors 4), MK (Megakaryocyte), Ery (Erythrocyte), Ma (Mast cell), Neu (neutrophil), Mo (monocyte), DC (dendritic cell), B cell and T cell, revealing dynamic changes in cellular composition across the recovery timeline (Figure 10A,B). Population dynamics analysis revealed HSPC (Hematopoietic Stem and Progenitor Cell) components, particularly HSCs, decreased dramatically after radiation, reaching their nadir around day 5, followed by gradual recovery (Figure 10D), while an erythroid contribution bias emerged after radiation, peaking at day 8 and persisting until day 21 (Figure 10E), and myeloid cell populations showed characteristic temporal changes throughout recovery (Figure 10F). Using pseudotime as a cellular primitivity metric, the recovery process demonstrated HSCs gradually regaining their primitive state with coordinated expression of primitive factors (*Hlf*, *Meis1*, *Satb1*, *Hoxa9*) indicating primitivity recovery, while myeloid factors (*Spi1*, *Gfi1*, *Cebpa*, *Cebpe*) and erythroid factors (*Klf1*, *Gata1*) exhibited lineage-specific temporal dynamics (Figure A9B and Figure 10C). scRL computation of lineage decision values for erythroid and myeloid lineages across all time points revealed characteristic spatial and temporal patterns projected onto UMAP space (Figure 10G,H), with comprehensive analysis showing distinct temporal patterns in both contribution and decision values (Figure A9C–E). Detailed analysis within HSPC subpopulations across uniformly 50-binned pseudotime demonstrated that erythroid-biased stem cell fate decision potential decreased dramatically immediately after radiation but rapidly returned to near-normal levels by day 8, while myeloid-biased stem cell proportions increased following irradiation and subsequently decreased as erythroid-biased populations recovered (Figure 10I,J). Correlation analysis revealed erythroid potential of HSPCs showed strongest association with overall HSPC proportions, whereas myeloid potential was closely correlated with myeloid cell proportions during early recovery phases, with validation through Pearson correlation coefficients confirming that myeloid and erythroid decision intensities exhibited positive correlations with their corresponding lineage-specific transcription factors while pseudotime demonstrated negative correlation with primitive transcription factor expression (Figure A9F and Figure 10K).
Figure 10Erythroid bias of primitive HSCs correlated with their proportion in time series of irradiation recovery. (GSE278673) (**A**) LSK phenotype bone marrow cells with clusters annotated as HSC (Hematopoietic Stem Cell), MPP2/3 (Multipotent Progenitors 2/3), MPP4 (Multipotent Progenitors 4), MK (Megakaryocyte), Ery (Erythrocyte), Ma (Mast cell), Neu (neutrophil), Mo (monocyte), DC (dendritic cell), B (B cell) and T (T cell). Different colors represent distinct cell subpopulations throughout the analysis. (**B**) Single-cell atlas of irradiation-injured LSK at time points 2, 5, 8, 11, 14, 21 and 30 days post-radiation, colored with annotated cell types. (**C**) Box plot of pseudotime distribution at each time point post-irradiation. (**D**) HSPC (Hematopoietic Stem and Progenitor Cell: HSC, MPP2/3, MPP4) proportion at each time point. (**E**) Erythroid cells (MK, Ery, Ma) proportion at each time point. (**F**) Myeloid cells (Neu, Mo, DC) proportion at each time point. (**G**) Erythroid lineage decision value projected on UMAP space at different time points. (**H**) Myeloid lineage decision value projected on UMAP space at different time points. (**I**) Lineage decision atlas for each time point within HSPC populations, showing erythroid decision values (red dots) and myeloid decision values (blue dots) along uniformly 50-binned pseudotime. (**J**) Horizontal bar plot of erythroid-biased proportion (red) and myeloid-biased proportion (blue) of HSPC at different time points. (**K**) Horizontal bar plot of Pearson correlations between erythroid-biased and myeloid-biased HSPC proportions at each time point with respect to average pseudotime, HSPC proportion, erythroid proportion and myeloid proportion.
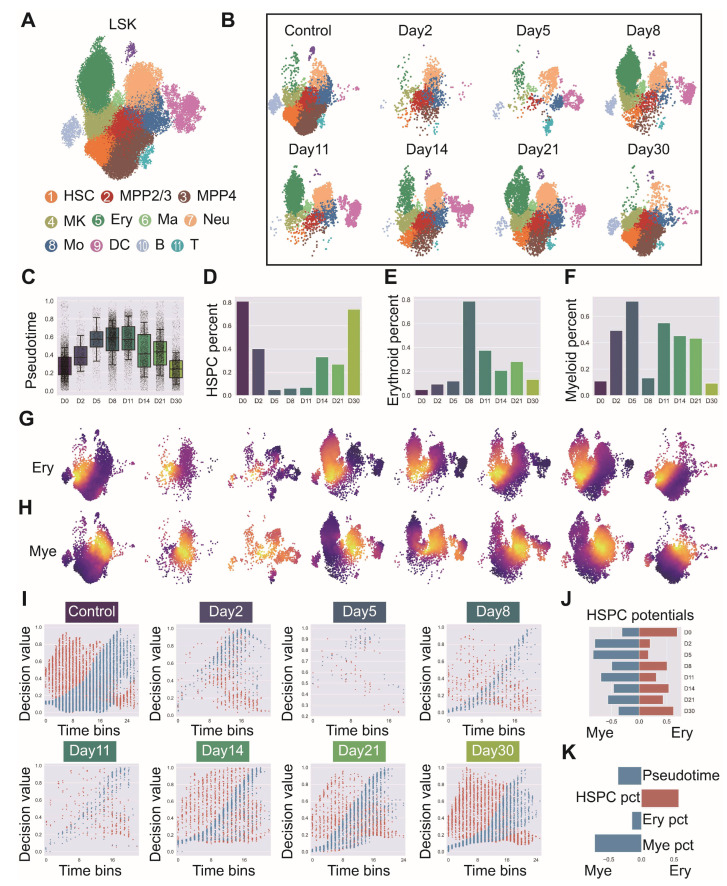



## 4. Discussion

The adaptation of reinforcement learning methodologies to biological systems represents a significant advancement in computational biology, addressing a fundamental gap in single-cell trajectory analysis: the identification of *where* and *when* fate decisions occur during cellular differentiation [45,46]. While traditional approaches primarily focus on ordering cells along developmental trajectories, scRL’s actor–critic framework recasts differentiation as a sequential decision process, enabling quantitative assessment of fate intensity and regulatory checkpoints that precede overt lineage commitment [32,33]. This paradigm shift from descriptive to mechanistic analysis represents a crucial step toward understanding the decision-making logic that governs cellular differentiation in both healthy and diseased contexts.

scRL’s superior performance across diverse biological systems—including human hematopoiesis, acute myeloid leukemia, mouse endocrinogenesis and perturbation studies—demonstrates its versatility in capturing early decision states that conventional methods fail to detect. By integrating LDA-derived interpretable latent spaces with biologically informed reward functions, scRL’s critic network learns state-value functions that quantify lineage potential, while the actor component traces optimal developmental routes across the manifold. This dual approach enables the framework to balance exploration of novel differentiation pathways with exploitation of established routes, consistently outperforming fifteen state-of-the-art methods across five independent evaluation dimensions. The identification of previously uncharacterized regulators such as *Dapp1* further validates scRL’s capacity to uncover biologically meaningful insights that extend beyond trajectory reconstruction.

A critical consideration in single-cell trajectory inference is the potential confounding effect of doublets—cellular aggregates that arise during droplet-based capture and may create spurious intermediate states with hybrid transcriptional profiles [47,48]. However, rigorous quality control measures implemented in preprocessing pipelines substantially mitigate doublet-induced artifacts, as these aggregates typically exhibit abnormally elevated transcript counts enabling their identification and removal through standard filtering criteria. Any residual doublets manifest as sporadic noise points rather than coherent cellular populations, appearing as isolated outliers in two-dimensional manifold representations that minimally impact the structural integrity of inferred developmental pathways.

The reliance on two-dimensional embeddings, while computationally tractable and visually interpretable, represents a pragmatic compromise between analytical precision and biological interpretability [49,50]. Although high-dimensional single-cell data exhibit complex nonlinear structure that may not be fully preserved in 2D projections, the interpretability advantages substantially outweigh potential inaccuracies introduced by dimensional compression. This strategic design choice enables meaningful biological discovery and experimental validation within a comprehensible framework that facilitates integration of computational predictions with experimental observations.

While scRL’s graph-based pseudotime alignment provides robust temporal ordering across grid representations, several methodological limitations merit consideration. The requirement for user-defined starting points introduces subjectivity that may bias trajectory inference, as suboptimal selection of early cells or clusters propagates errors throughout Dijkstra’s shortest-path calculations [51]. More fundamentally, the assumption that shortest-path distances correspond to developmental time may not accurately reflect cellular differentiation dynamics, particularly in systems with rapid state transitions or non-monotonic developmental patterns [52,53]. Grid discretization compounds these issues by introducing artifacts through rasterization of continuous UMAP embeddings, potentially creating artificial barriers or connections that misrepresent the underlying biological manifold [54]. The multi-component handling strategy, despite its mathematical elegance, introduces approximation errors when connecting disconnected graph components, potentially mischaracterizing relationships between separated cellular populations [55]. Finally, the static, undirected graph structure overlooks the inherently directional nature of developmental processes, while the absence of uncertainty quantification limits reliability assessment, particularly in sparsely sampled regions where multiple plausible trajectories may exist [56].

The selection of grid embedding hyperparameters *n* and *j* critically influences grid construction and computational efficiency. Based on systematic parameter evaluation across grid resolution *n* (25, 50, 75, 100, 125) and border observer parameter *j* (3, 5, 8, 11, 15), we recommend n=50 and j=3 as optimal choices that balance embedding accuracy with computational tractability (Figure A10).

Beyond trajectory inference, scRL’s unified framework addresses broader challenges in developmental biology by providing competitive measures of lineage-contribution intensity without requiring ground-truth probabilities. This capability, combined with its ability to reconcile gene- and lineage-level information across healthy, diseased and perturbed systems, positions scRL as a versatile tool for regenerative medicine, oncology and developmental biology. The framework’s extensibility toward multi-omic and spatial modalities, coupled with ongoing efforts to streamline unsupervised identification of key subpopulations, promises to advance our mechanistic understanding of cell fate decisions and their therapeutic manipulation.

## 5. Conclusions

scRL brings reinforcement learning into single-cell biology, offering a principled, data-driven approach for charting differentiation landscapes and pinpointing regulatory checkpoints. Its ability to capture early decision states, reconcile gene- and lineage-level information, and generalize across healthy, diseased and perturbed systems positions scRL as a versatile tool for developmental biology, oncology and regenerative medicine. Ongoing work aims to integrate multi-omic and spatial modalities and to streamline unsupervised identification of key subpopulations, extending scRL’s reach toward a comprehensive, mechanistic atlas of cell fate decisions.

## Figures and Tables

**Table 1 biology-14-00679-t001:** LDA performance improvements over baseline methods across HVG sizes.

Method	NMI	ARI	ASW	DB	CH	Overall
LDA vs. scVI	+0.267	+0.199	+0.334	+1.504	+2151.767	+0.785
LDA vs. ICA	+0.220	+0.149	+0.331	+1.938	+1857.164	+0.675
LDA vs. PCA	+0.064	+0.058	+0.303	+1.316	+1745.558	+0.511
LDA vs. LscVI	+0.081	+0.032	+0.246	+1.213	+1786.359	+0.459
LDA vs. FA	+0.012	−0.021	+0.190	+0.557	+1577.247	+0.261
LDA vs. Diff	−0.016	−0.045	+0.057	+0.585	+1568.978	+0.204
LDA vs. NMF	+0.007	+0.010	+0.159	+0.179	+375.229	+0.123

**Table 2 biology-14-00679-t002:** LDA performance improvements over baseline methods across latent space sizes.

Method	NMI	ARI	ASW	DB	CH	Overall
LDA vs. scVI	+0.213	+0.106	+0.338	+1.516	+2643.360	+0.713
LDA vs. ICA	+0.127	+0.029	+0.334	+2.147	+2337.782	+0.618
LDA vs. LscVI	+0.075	+0.028	+0.326	+1.451	+2354.817	+0.560
LDA vs. PCA	−0.003	−0.056	+0.293	+1.576	+2273.306	+0.409
LDA vs. FA	−0.005	−0.084	+0.260	+0.686	+2383.781	+0.278
LDA vs. Diff	−0.042	−0.099	+0.124	+0.796	+2157.947	+0.198
LDA vs. NMF	−0.066	−0.117	+0.153	+0.536	+886.494	+0.075

**Table 3 biology-14-00679-t003:** scRL performance improvements over LDA percentile variants on hematopoietic dataset.

Method	ARI	NMI	ASW	CH	DB	Overall
scRL vs. ${\mathrm{LDA}}_{80\%}$	+0.330	+0.209	+0.549	+1016.722	+0.495	+0.986
scRL vs. ${\mathrm{LDA}}_{85\%}$	+0.308	+0.192	+0.533	+981.563	+0.465	+0.932
scRL vs. ${\mathrm{LDA}}_{90\%}$	+0.253	+0.170	+0.434	+851.186	+0.350	+0.735
scRL vs. ${\mathrm{LDA}}_{95\%}$	+0.153	+0.091	+0.251	+482.741	+0.172	+0.375

**Table 4 biology-14-00679-t004:** scRL fate decision performance improvements over baseline methods on hematopoietic dataset.

Method	NMI	ARI	ASW	DB	CH	Overall
scRL vs. ${\mathrm{Diff}}_{80\%}$	+0.121	+0.060	+0.224	+1.314	+966.124	+0.465
scRL vs. ${\mathrm{Diff}}_{85\%}$	+0.115	+0.046	+0.213	+1.265	+940.620	+0.441
scRL vs. ${\mathrm{Diff}}_{90\%}$	+0.100	+0.039	+0.256	+0.939	+927.518	+0.432
scRL vs. ${\mathrm{Diff}}_{95\%}$	+0.037	−0.007	+0.103	+0.638	+720.043	+0.228
scRL vs. ${\mathrm{FA}}_{80\%}$	+0.229	+0.142	+0.226	+2.317	+979.821	+0.647
scRL vs. ${\mathrm{FA}}_{85\%}$	+0.195	+0.114	+0.202	+1.892	+926.712	+0.563
scRL vs. ${\mathrm{FA}}_{90\%}$	+0.150	+0.076	+0.233	+1.090	+836.652	+0.466
scRL vs. ${\mathrm{FA}}_{95\%}$	+0.080	+0.009	+0.186	+0.578	+648.564	+0.296
scRL vs. ${\mathrm{ICA}}_{80\%}$	+0.346	+0.224	+0.243	+3.010	+995.896	+0.811
scRL vs. ${\mathrm{ICA}}_{85\%}$	+0.344	+0.224	+0.241	+2.829	+968.163	+0.785
scRL vs. ${\mathrm{ICA}}_{90\%}$	+0.325	+0.213	+0.231	+2.577	+926.928	+0.731
scRL vs. ${\mathrm{ICA}}_{95\%}$	+0.258	+0.162	+0.206	+2.548	+837.487	+0.614
scRL vs. ${\mathrm{LscVI}}_{80\%}$	+0.236	+0.170	+0.262	+1.989	+1014.079	+0.677
scRL vs. ${\mathrm{LscVI}}_{85\%}$	+0.213	+0.155	+0.254	+1.674	+991.179	+0.627
scRL vs. ${\mathrm{LscVI}}_{90\%}$	+0.169	+0.125	+0.239	+1.365	+952.584	+0.551
scRL vs. ${\mathrm{LscVI}}_{95\%}$	+0.119	+0.078	+0.208	+1.228	+883.328	+0.454
scRL vs. ${\mathrm{NMF}}_{80\%}$	+0.307	+0.262	+0.008	+0.095	+656.817	+0.455
scRL vs. ${\mathrm{NMF}}_{85\%}$	+0.311	+0.258	+0.093	+0.143	+817.645	+0.540
scRL vs. ${\mathrm{NMF}}_{90\%}$	+0.278	+0.237	+0.158	+0.169	+799.054	+0.544
scRL vs. ${\mathrm{NMF}}_{95\%}$	+0.160	+0.171	+0.096	+0.023	+349.374	+0.297
scRL vs. ${\mathrm{PCA}}_{80\%}$	+0.240	+0.166	+0.199	+2.434	+920.403	+0.651
scRL vs. ${\mathrm{PCA}}_{85\%}$	+0.206	+0.150	+0.179	+2.095	+861.993	+0.579
scRL vs. ${\mathrm{PCA}}_{90\%}$	+0.152	+0.110	+0.156	+1.697	+792.187	+0.474
scRL vs. ${\mathrm{PCA}}_{95\%}$	+0.089	+0.059	+0.159	+1.205	+702.267	+0.362
scRL vs. ${\mathrm{scVI}}_{80\%}$	+0.344	+0.226	+0.200	+1.731	+1061.220	+0.733
scRL vs. ${\mathrm{scVI}}_{85\%}$	+0.325	+0.216	+0.195	+1.542	+1040.949	+0.698
scRL vs. ${\mathrm{scVI}}_{90\%}$	+0.299	+0.202	+0.190	+1.390	+1020.921	+0.659
scRL vs. ${\mathrm{scVI}}_{95\%}$	+0.259	+0.181	+0.185	+1.272	+998.612	+0.609

**Table 5 biology-14-00679-t005:** scRL lineage contribution performance improvements over baseline methods across different cluster numbers on hematopoietic dataset.

Method	ASW	DB	CH	Overall
scRL vs. ${\mathrm{PCA}}_{4}$	+0.247	+0.359	+5121.830	+0.353
scRL vs. ${\mathrm{ICA}}_{4}$	+0.256	+0.447	+6269.940	+0.409
scRL vs. ${\mathrm{FA}}_{4}$	+0.119	+0.100	+5613.054	+0.242
scRL vs. ${\mathrm{NMF}}_{4}$	+0.114	+0.105	+816.813	+0.105
scRL vs. ${\mathrm{Diff}}_{4}$	+0.041	+0.012	−2261.901	−0.048
scRL vs. ${\mathrm{scVI}}_{4}$	+0.551	+1.761	+8639.039	+0.899
scRL vs. ${\mathrm{LscVI}}_{4}$	+0.241	+0.350	+4669.743	+0.336
scRL vs. ${\mathrm{PCA}}_{6}$	+0.124	+0.235	+5058.890	+0.329
scRL vs. ${\mathrm{ICA}}_{6}$	+0.150	+0.312	+6535.205	+0.425
scRL vs. ${\mathrm{FA}}_{6}$	+0.058	+0.023	+5792.529	+0.254
scRL vs. ${\mathrm{NMF}}_{6}$	+0.042	+0.077	+2848.851	+0.131
scRL vs. ${\mathrm{Diff}}_{6}$	−0.086	+0.212	+2454.264	−0.024
scRL vs. ${\mathrm{scVI}}_{6}$	+0.368	+1.301	+7855.647	+0.859
scRL vs. ${\mathrm{LscVI}}_{6}$	+0.119	+0.255	+5366.970	+0.337
scRL vs. ${\mathrm{PCA}}_{8}$	+0.157	+0.310	+3963.771	+0.426
scRL vs. ${\mathrm{ICA}}_{8}$	+0.189	+0.353	+5198.874	+0.535
scRL vs. ${\mathrm{FA}}_{8}$	+0.120	+0.080	+4634.559	+0.375
scRL vs. ${\mathrm{NMF}}_{8}$	+0.103	+0.196	+2651.048	+0.265
scRL vs. ${\mathrm{Diff}}_{8}$	−0.047	+0.133	+3041.397	+0.091
scRL vs. ${\mathrm{scVI}}_{8}$	+0.360	+1.106	+5999.717	+0.924
scRL vs. LscVI_8_	+0.177	+0.320	+4190.457	+0.456
scRL vs. PCA_10_	+0.137	+0.237	+3330.795	+0.379
scRL vs. ICA_10_	+0.169	+0.280	+3284.407	+0.430
scRL vs. FA_10_	+0.069	+0.112	+3547.165	+0.236
scRL vs. NMF_10_	+0.061	+0.103	+2414.816	+0.210
scRL vs. Diff_10_	−0.069	+0.161	+2894.393	+0.059
scRL vs. scVI_10_	+0.308	+1.024	+4944.921	+0.856
scRL vs. ${\mathrm{LscVI}}_{10}$	+0.161	+0.321	+3663.236	+0.447
scRL vs. PCA_12_	+0.194	+0.465	+2821.714	+0.515
scRL vs. ICA_12_	+0.235	+0.578	+3874.281	+0.694
scRL vs. FA_12_	+0.096	+0.106	+2951.559	+0.284
scRL vs. NMF_12_	+0.114	+0.240	+2419.564	+0.334
scRL vs. Diff_12_	−0.064	+0.043	+2708.683	+0.120
scRL vs. scVI_12_	+0.300	+1.019	+4055.950	+0.899
scRL vs. LscVI_12_	+0.164	+0.311	+2998.675	+0.461

## Data Availability

The single-cell RNA-seq data have been deposited in the GEO database under accession numbers GSE277292 and GSE278673 and are publicly available. All original codes have been deposited at GitHub (v0.0.5) at https://github.com/PeterPonyu/scRL accessed on 19 November 2024.

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
