# Peer review of "scRL: Utilizing Reinforcement Learning to Evaluate Fate Decisions in Single-Cell Data"

_biology, 2025, doi:10.3390/biology14060679_

Round 1
Reviewer 1 Report
Comments and Suggestions for Authors
This manuscript introduces scRL, a reinforcement learning (RL)–based framework designed to model cell fate decisions in single-cell RNA-seq datasets. The authors implement an actor-critic algorithm to simulate trajectories within a grid-based embedding and quantify “decision intensity” associated with gene- and lineage-level differentiation. The framework is applied across multiple datasets, with claims of enhanced interpretability and improved trajectory modeling.
While the conceptual framework is novel and holds potential, the manuscript exhibits several critical weaknesses in figure clarity, methodological rigor, biological validation, and comparative benchmarking. These shortcomings limit the accessibility and evaluative strength of the work.
- The manuscript contains an excessive number of complex, multi-panel figures in the main text. Many are poorly annotated, difficult to interpret, or redundant. This significantly impacts the clarity and accessibility of the paper.
Using Figure 2 as an example below and the authors really need to work more on all figures-
• Figure 2A: Terms such as “Ngn3 low EP,” “Fev+,” “alpha,” “beta,” and “delta” are used without explanation. Full cell type names should be introduced at first mention.
• Figure 2B: The rationale for displaying only a limited set of marker genes is unclear. This actually affects the interpretability of subsequent panels (2C/2D). Consider moving full marker gene visualizations to supplementary figures.
• Figures 2C–2E: The grid structure and heatmap color scales are not explained; color bars are missing.
• Figures 2F–2G: Axes lack description. It is unclear whether these represent pseudotime bins, grid indices, or PCA components.
• Figure 2L: Terms such as “Decision” and “Contribution” are not defined in the figure or text.
• Across all panels: Gene expression values are shown without clarifying whether they are normalized, log-transformed, or scaled. - The authors benchmark scRL primarily against dimensionality reduction methods (e.g., LDA, scVI, PHATE), but do not include comparisons with well-established fate inference tools such as CellRank or CellOracle. These are directly relevant and widely used for trajectory modeling and fate probability estimation.
- The manuscript does not mention or address the presence of doublets or multiplets, which can significantly distort trajectory inference and embedding quality. These artifacts may produce spurious intermediate states, particularly problematic for a model that quantifies decision-making intensity.
- Several key components of the scRL framework are not clearly described. In particular, the grid embedding procedure lacks a formal definition. It is unclear how the grid is constructed, how resolution is chosen, and how cells are mapped to it. The reward function and actor-critic policy design are described only qualitatively. Hyperparameter selection is also not addressed in detail.
- Figures 2–6 present overlapping content using the same datasets — such as gene expression patterns, trajectory mappings, and decision metrics. This redundancy weakens the manuscript's focus and contributes to figure overload. The authors could streamline the figures by consolidating overlapping results. Use one dataset as a main demonstration and move supporting examples to supplementary materials.
- The manuscript contains awkward phrasing and uses technical jargon that may be inaccessible to readers without a strong computational background. Phrases like “superior interpretability” and “yield a decision mode that diminishes…” are vague. Abbreviations (e.g., MEP, GMP, EP) were not spelled out at first mention.
- Define “decision intensity” clearly and early in the manuscript.
- Improve figure legends by including axis labels, units, and color annotations.
- Clarify which pseudotime inference method is used and discuss its assumptions and limitations.
- Expand citations to include key fate inference tools (e.g., CellRank, Palantir).
- Italicize all gene symbols consistently.
- Figure A1 legend is incomplete and should be revised.
Author Response
Reviewer #1
Comments and Suggestions for Authors:
This manuscript introduces scRL, a reinforcement learning (RL)–based framework designed to model cell fate decisions in single-cell RNA-seq datasets. The authors implement an actor-critic algorithm to simulate trajectories within a grid-based embedding and quantify “decision intensity” associated with gene- and lineage-level differentiation. The framework is applied across multiple datasets, with claims of enhanced interpretability and improved trajectory modeling.
While the conceptual framework is novel and holds potential, the manuscript exhibits several critical weaknesses in figure clarity, methodological rigor, biological validation, and comparative benchmarking. These shortcomings limit the accessibility and evaluative strength of the work.
#Comments. 1
The manuscript contains an excessive number of complex, multi-panel figures in the main text. Many are poorly annotated, difficult to interpret, or redundant. This significantly impacts the clarity and accessibility of the paper.
Using Figure 2 as an example below and the authors really need to work more on all figures-
• Figure 2A: Terms such as “Ngn3 low EP,” “Fev+,” “alpha,” “beta,” and “delta” are used without explanation. Full cell type names should be introduced at first mention.
• Figure 2B: The rationale for displaying only a limited set of marker genes is unclear. This actually affects the interpretability of subsequent panels (2C/2D). Consider moving full marker gene visualizations to supplementary figures.
• Figures 2C–2E: The grid structure and heatmap color scales are not explained; color bars are missing.
• Figures 2F–2G: Axes lack description. It is unclear whether these represent pseudotime bins, grid indices, or PCA components.
• Figure 2L: Terms such as “Decision” and “Contribution” are not defined in the figure or text.
• Across all panels: Gene expression values are shown without clarifying whether they are normalized, log-transformed, or scaled.
#Response. 1
We thank the reviewer for the valuable feedback regarding figure complexity and annotation. In response, we have substantially revised all figures to improve clarity and readability. We simplified multi-panel layouts, clarified all abbreviations and cell type names, and added detailed explanations for gene selection, grid structures, heatmap color scales, and axes. Missing color bars and scale indicators have been included, and all data transformations are now clearly labeled. (Figure 1-10, Figure A1-10, Table 1-5, Table B1-5)
#Comments. 2
The authors benchmark scRL primarily against dimensionality reduction methods (e.g., LDA, scVI, PHATE), but do not include comparisons with well-established fate inference tools such as CellRank or CellOracle. These are directly relevant and widely used for trajectory modeling and fate probability estimation.
#Response. 2
We thank the reviewer for highlighting the importance of benchmarking against established fate inference tools. In the revised manuscript, we have included comprehensive comparisons between scRL and widely-used fate inference methods such as CellRank, Palantir, and FateID. (Figure 6, Figure A6, p. 22, line 517-532)
#Comments. 3
The manuscript does not mention or address the presence of doublets or multiplets, which can significantly distort trajectory inference and embedding quality. These artifacts may produce spurious intermediate states, particularly problematic for a model that quantifies decision-making intensity.
#Response. 3
We thank the reviewer for raising this important concern. Doublets and multiplets can indeed introduce spurious intermediate states that potentially affect trajectory inference and the quantification of decision-making intensity. In our workflow, we mitigate these artifacts through rigorous quality control during data preprocessing, as doublets generally present with abnormally high transcript and gene counts and are removed using standard filtering criteria. Any residual doublets that are not excluded typically appear as isolated noise points rather than forming coherent cell populations, and thus are unlikely to systematically distort trajectory reconstruction. Additionally, in two-dimensional manifold representations, these remaining doublets would not dominate the main cellular states, minimizing their impact on the integrity of inferred developmental pathways. (p. 31, line 677-685)
#Comments. 4
Several key components of the scRL framework are not clearly described. In particular, the grid embedding procedure lacks a formal definition. It is unclear how the grid is constructed, how resolution is chosen, and how cells are mapped to it. The reward function and actor-critic policy design are described only qualitatively. Hyperparameter selection is also not addressed in detail.
#Response. 4
We thank the reviewer for highlighting the need for greater methodological clarity. In response, we have substantially revised the Methods and Materials section to include comprehensive technical details and formal mathematical definitions. We now provide step-by-step descriptions of grid construction, resolution selection, cell-to-grid mapping, the reward function, and the actor-critic policy architecture. (p. 3-17, line 87-437) Hyperparameter selection is discussed with additional results (p. 32, line 711-715, Figure A10)
#Comments. 5
Figures 2–6 present overlapping content using the same datasets — such as gene expression patterns, trajectory mappings, and decision metrics. This redundancy weakens the manuscript's focus and contributes to figure overload. The authors could streamline the figures by consolidating overlapping results. Use one dataset as a main demonstration and move supporting examples to supplementary materials.
#Response. 5
We thank the reviewer for this important suggestion. In response, we have comprehensively reorganized Figures 2–6 by removing redundant panels and consolidating overlapping results. As recommended, we now use one primary dataset for the main demonstration in the core figures to improve clarity and narrative focus. Supporting results and examples from additional datasets have been moved to the supplementary materials, where they remain accessible for validation. (Figure 2-6, Figure A2-6, Table 1-5, Table B1-5)
#Comments. 6
The manuscript contains awkward phrasing and uses technical jargon that may be inaccessible to readers without a strong computational background. Phrases like “superior interpretability” and “yield a decision mode that diminishes…” are vague. Abbreviations (e.g., MEP, GMP, EP) were not spelled out at first mention.
#Response. 6
We appreciate the reviewer’s feedback regarding language clarity and the use of specialized terminology. In response, we have carefully revised the manuscript to improve the precision and objectivity of our results descriptions, ensuring that all statements are explicitly supported by data rather than subjective interpretation. Additionally, we have provided the full names for all abbreviations (such as megakaryocyte-erythroid progenitor [MEP], granulocyte-monocyte progenitor [GMP], and endocrinogenesis progenitor [EP]) at their first occurrence in the text.
#Comments. 7
Define “decision intensity” clearly and early in the manuscript.
#Response. 7
We thank the reviewer for highlighting the need for a clear definition of “decision intensity.” In our revised manuscript, we explicitly define decision intensity in the Abstract to provide early clarity, and further elaborate on its specific meaning and implementation within the scRL framework in the Methods and Materials section. (p. 4, line 115-129)
#Comments. 8
Improve figure legends by including axis labels, units, and color annotations.
#Response. 8
We appreciate the reviewer’s suggestion regarding the clarity of figure legends. We have reconstructed nearly all figures and their captions to ensure that key results are easily interpretable for readers.
#Comments. 9
Clarify which pseudotime inference method is used and discuss its assumptions and limitations.
#Response. 9
We thank the reviewer for this important point. In the revised manuscript, we clearly specify the pseudotime inference method utilized in the Methods and Materials section, providing relevant methodological details. Furthermore, we have added a discussion of its underlying assumptions and limitations in the Discussion section to enhance transparency and contextual understanding for readers. (p. 7, line 174-198, p. 31-32, line 694-710)
#Comments. 10
Expand citations to include key fate inference tools (e.g., CellRank, Palantir).
#Response. 10
We appreciate the reviewer’s suggestion. In our revised manuscript, we have expanded the citations to include all key fate inference tools, such as CellRank and Palantir, when making comparisons with these methods. (p. 22, line 517-532)
#Comments. 11
Italicize all gene symbols consistently.
#Response. 11
We thank the reviewer for their attention to formatting consistency. In the revised manuscript, all gene symbols are now consistently italicized throughout the text.
#Comments. 12
Figure A1 legend is incomplete and should be revised.
#Response. 12
We appreciate the reviewer’s feedback regarding Figure A1. We have revised and expanded the legend for Figure A1 to ensure it is complete and informative. Additionally, we have reconstructed most of the results figures to enhance their clarity and readability for the reader.
Reviewer 2 Report
Comments and Suggestions for Authors
Authors presented a new computational tool for single-cell data analysis that implements machine-learning methods utilizing reinforcement learning. They used this tool to assess cell fate decisions and track differentiation trajectories on six publicly available datasets. They demonstrated that their method mostly outperforms other known computational methods on these datasets in various aspects of the problem under study. The manuscript is well written, although with a balance shifted towards technical aspects rather than the biological results. The conclusions are well supported by the results. Overall, I believe the presented tool will find applications and further development in the research community associated with the single-cell data analysis. Therefore, I have only minor comments/questions regarding the manuscript:
- The manuscript is highly technical, so it would be extremely useful to start with an illustrative figure with an example of how the presented method uses reinforcement while solving the problems of tracking differentiation trajectories and cell fate decisions. This figure should be understandable for a wider audience, beyond the machine-learning community.
- As I understand, the 2D embedding space is used for the convenience of visualization, but I presume for the analysis the dimensionality of the latent space should not play a crucial role. What if the 2D representation is not enough for the data under analysis, would it be possible to apply the whole approach for a higher dimensional embedding space?
- Line 368 and similar lines in Methods: More comments reminding the definition of the parameters n and j are required here.
- Figure 3 caption and similar figures: Each figure with the results concerning a specific dataset should mention this dataset.
Author Response
Reviewer #2
Comments and Suggestions for Authors:
Authors presented a new computational tool for single-cell data analysis that implements machine-learning methods utilizing reinforcement learning. They used this tool to assess cell fate decisions and track differentiation trajectories on six publicly available datasets. They demonstrated that their method mostly outperforms other known computational methods on these datasets in various aspects of the problem under study. The manuscript is well written, although with a balance shifted towards technical aspects rather than the biological results. The conclusions are well supported by the results. Overall, I believe the presented tool will find applications and further development in the research community associated with the single-cell data analysis. Therefore, I have only minor comments/questions regarding the manuscript:
#Comments. 1
The manuscript is highly technical, so it would be extremely useful to start with an illustrative figure with an example of how the presented method uses reinforcement while solving the problems of tracking differentiation trajectories and cell fate decisions. This figure should be understandable for a wider audience, beyond the machine-learning community.
#Response. 1
We appreciate the reviewer’s suggestion. We have added a graphical abstract and key points to the revised manuscript to clearly illustrate our study’s logic and innovations. (p. 2)
#Comments. 2
As I understand, the 2D embedding space is used for the convenience of visualization, but I presume for the analysis the dimensionality of the latent space should not play a crucial role. What if the 2D representation is not enough for the data under analysis, would it be possible to apply the whole approach for a higher dimensional embedding space?
#Response. 2
We thank the reviewer for raising this insightful question regarding the dimensionality of the embedding space. In the revised manuscript, we have expanded the Discussion section to address the appropriateness of two-dimensional embeddings for our analysis. (p. 31, line 686-693)
#Comments. 3
Line 368 and similar lines in Methods: More comments reminding the definition of the parameters n and j are required here.
#Response. 3
We thank the reviewer for this helpful suggestion. In the revised Methods section, we have added explicit definitions of the parameters $n$ and $j$, as well as explanations of their roles within scRL, to improve clarity for readers. (p. 12-13, line 298-302)
#Comments. 4
Figure 3 caption and similar figures: Each figure with the results concerning a specific dataset should mention this dataset.
#Response. 4
We thank the reviewer for this suggestion. We have updated all relevant figure captions to specify the corresponding datasets, including information on public access and appropriate references. (Figure 2-10, Figure A1-9)
Round 2
Reviewer 1 Report
Comments and Suggestions for Authors
The revised manuscript has improved.